# Advantages of eutectic alloys for creating catalysts in the realm of nanotechnology-enabled metallurgy

Jianbo Tang [1], Rahman Daiyan[1], Mohammad B. Ghasemian [1], Shuhada A. Idrus-Saidi[1], Ali Zavabeti[2,3], Torben Daeneke[2], Jiong Yang[1], Pramod Koshy[4], Soshan Cheong [5], Richard D. Tilley [5,6,7], Richard B. Kaner[8,9], Rose Amal[1] & Kourosh Kalantar-Zadeh [1]*

The nascent field of nanotechnology-enabled metallurgy has great potential. However, the role of eutectic alloys and the nature of alloy solidification in this field are still largely unknown. To demonstrate one of the promises of liquid metals in the field, we explore a model system of catalytically active Bi-Sn nano-alloys produced using a liquid-phase ultra-sonication technique and investigate their phase separation, surface oxidation, and nucleation. The Bi-Sn ratio determines the grain boundary properties and the emergence of dislocations within the nano-alloys. The eutectic system gives rise to the smallest grain dimensions among all Bi-Sn ratios along with more pronounced dislocation formation within the nano-alloys. Using electrochemical $CO_2$ reduction and photocatalysis, we demonstrate that the structural peculiarity of the eutectic nano-alloys offers the highest catalytic activity in comparison with their non-eutectic counterparts. The fundamentals of nano-alloy formation revealed here may establish the groundwork for creating bimetallic and multimetallic nano-alloys.

[1] School of Chemical Engineering, University of New South Wales (UNSW), Sydney, NSW 2052, Australia. [2] School of Engineering, RMIT University, Melbourne, VIC 3001, Australia. [3] College of Material Science and Technology, Nanjing University of Aeronautics and Astronautics, 29 Jiangjun Ave, 211100 Nanjing, China. [4] School of Materials Science and Engineering, UNSW, Sydney, NSW 2052, Australia. [5] Mark Wainwright Analytical Centre, UNSW, Sydney, NSW 2052, Australia. [6] School of Chemistry, UNSW, Sydney, NSW 2052, Australia. [7] Australian Centre for NanoMedicine, UNSW, Sydney, NSW 2052, Australia. [8] Department of Chemistry and Biochemistry, University of California, Los Angeles (UCLA), Los Angeles, CA 90095, USA. [9] Department of Materials Science and Engineering, UCLA, Los Angeles, CA 90095, USA. *email: k.kalantar-zadeh@unsw.edu.au

It is an ancient wisdom that alloys produced by mixing different metals together, such as bronze, brass, and pewter, can offer more desirable properties than their single-metal counterparts[1]. Interests in alloys have not subsided in present time but rather significantly expanded as combinatoric metallurgy in pursuit of well-tailored alloys. These efforts have led to the discovery of metallic glasses[2], ultrahigh strength super alloys[3], high-entropy alloys[4], and many more[5–7].

Interesting behaviours are seen in liquid metal alloys. It has been known for long that mixing different metals can lead to a decrease in both melting and freezing temperatures[8]. At an exact mixing ratio, the temperature drop becomes most significant, leading to the state referred to as the eutectic, from the Greek word meaning 'easily melting'. The single-phase-like transition behaviour of eutectic systems has been shown to be advantageous for many technologically important applications, such as heat exchange and electronic switches[9–11]. While these applications mostly concern liquid metals in their bulk form, in recent years more knowledge regarding the properties in low dimensions, has been gained, thanks to advancements in electron microscopy and other spectroscopic capabilities[12].

To explore the fundamentals of liquid alloys across different scales, both their surface and core should be studied. To date, such studies on bulk systems have resulted in intriguing observations. For instance, despite the fact that liquid metals and some of their alloys are defined by their disordered condensed state within their bulk, X-ray reflectivity analyses have revealed that their surfaces are ordered at atomic level[13–15], a unique virtue of liquid alloys which has led to new methods for creating two-dimensional materials[16,17]. Moreover, in a liquid metal bulk, adding extra metallic entities have been found to improve their catalysis performance[18]. Similarly, it is perceived that many other distinct traits of liquid metals can be observed in nanoscales.

From a scientific point of view, the solid-to-liquid transition is a game-changing characteristic of alloys, considering their unique transformations during solidification. By changing the conditions during mixing and solidification, interesting phase separation can be achieved within the alloys and unique crystal phases can be developed[19,20], depending on the nature of the liquid metal mixture and the applied stimuli. One of the stimuli is mechanical that can be applied by means of ultrasonication. Fundamentally, ultrasonication induces high intensity shear stress by ultrasonic irradiation to achieve fragmentation of liquid specimen[21]. It is a high-throughput, readily scalable method for the synthesis of micro-material and nano-material from liquid metals[22–24]. While ultrasonication cannot be directly applied to many metals individually due to their high melting temperature, the melting point drop via mixing different metals can make their alloys accessible for ultrasonication processing. Therefore, combining ultrasonication with traditional metallurgy practices could be a potential strategy to realise liquid metal pathways for scalable synthesis of high value microalloys and nano-alloys, which could be promising for catalysis, optoelectronics, and biodiagnostics[25–28].

In this work, we synthesise nano-alloys by ultrasonication and investigate the intrinsic phase separation within the nanoparticles during solidification. Ultrasonication process is pursued for creating catalytic nano-alloys as it offers high yield, low cost, and avoids unwanted by-products that are formed in many typical chemical/electrochemical reactions[29,30]. Many multimetallic nano-alloys are known to be efficient catalysts relative to their monometallic counterparts[25,26], so here as a proof-of-concept, we select bismuth–tin (Bi–Sn) alloys as the input for our system. Advantageously, these binary alloys show no intermetallic phase, which make the study less complex and more informative

(Fig. 1)[31]. As illustrated in Fig. 1a, we start by synthesising Bi–Sn bulk alloys for studying the phase separation, under no applied stimuli, in order to elucidate the crystal formation during the solidification at different Bi–Sn ratios. Liquid phase ultrasonication of these liquid bulk samples is then performed at an elevated temperature, which breaks the bulk samples into nano-alloys. We investigate changes in the structure and the phase-transition behaviour of these nano-alloys that arise from compositional differences. We compare the nano-alloys with their bulks, and examine atomic scale structures and defect formations, especially in the eutectic phase. Bi–Sn nano-alloys of different ratios are then investigated for photocatalysis degradation of organic pollutants (Fig. 1b) and electrochemical catalytic reduction of $CO_2$ (Fig. 1c). We highlight and discuss why the sample with the eutectic composition offers the best performance in the catalytic processes.

## Results

**Bi–Sn bulk alloys**. The study of the phase separation after solidification of liquid metal alloys in bulk form can give good indications on what may occur in nano-alloys. Here $Bi_xSn_{1-x}$ bulk alloys with Bi weight ratios of $x = 0.20$ (hypoeutectic), 0.40 (hypoeutectic), 0.57 (eutectic), 0.80 (hypereutectic), and the rest comprised of Sn, together with individual Sn ($x = 0.00$) and Bi ($x = 1.00$) metals, were prepared and characterised (see the "Methods" section). Ambient-temperature and non-directionally solidified Bi–Sn bulk alloys were fractured after liquid nitrogen cooling and the microstructures of their cross-sections were investigated using scanning electron microscopy (SEM) and energy-dispersive X-ray spectroscopy (EDX). Their transverse sections show microscale subdivisions of finely distributed rod-like, fibrous and lamellar phases (Fig. 2a–c, Supplementary Fig. 1). The change in Bi–Sn ratios gives rise to distinct morphologies, which are discussed below.

As shown in Fig. 2a–c, for the Bi–Sn sample with the lowest Bi ratio ($x = 0.20$), Bi forms micron/sub-micron dimensional rods (almost 100%), which discretely embedded into the Sn background phase. While with lower frequency of occurrence (16%), this discrete growth is also observed as the Bi ratio is increased to $x = 0.40$, where the Bi and Sn phases start to cut into each other, and the well-known fibrous and lamellar structures emerge[32]. When the Bi–Sn ratio reaches the eutectic value ($x = 0.57$), the fibrous (59%) and lamellar (41%) structures become dominant, ruling out the discrete growth regime as observed in the hypoeutectic samples. When the mixing becomes hypereutectic ($x = 0.80$), Bi grows into thick lamellae (76%) with small inter-lamella spacings.

The separation of the Bi and Sn phases inside the bulk samples can be understood given the fact that nucleation and the subsequent crystal growth of individual elements lead to phase redistribution during solidification[32,33]. We note that, either being fibrous or lamellar, the distribution of the Bi and Sn phases inside the eutectic sample is more uniform than the non-eutectics. In addition, the significantly distorted fibrous structures indicate stress build-up in the samples and therefore, the dominance of the fibrous structures in the eutectic sample implies its enhanced stress after solidification. Such structures of the eutectic sample have a thermodynamic origin, which is revealed by differential scanning calorimetry (DSC). As can be seen from the DSC curves in Fig. 2d, the eutectic sample ($x = 0.57$) is the only Bi–Sn mixture with a single endothermic peak (melting onset 139 °C) during the melting half cycle and a single exothermic peak during the other solidification half, which is by definition the phase transition behaviour of a eutectic system. Conversely, all non-eutectic mixtures show an endothermic peak at the eutectic onset

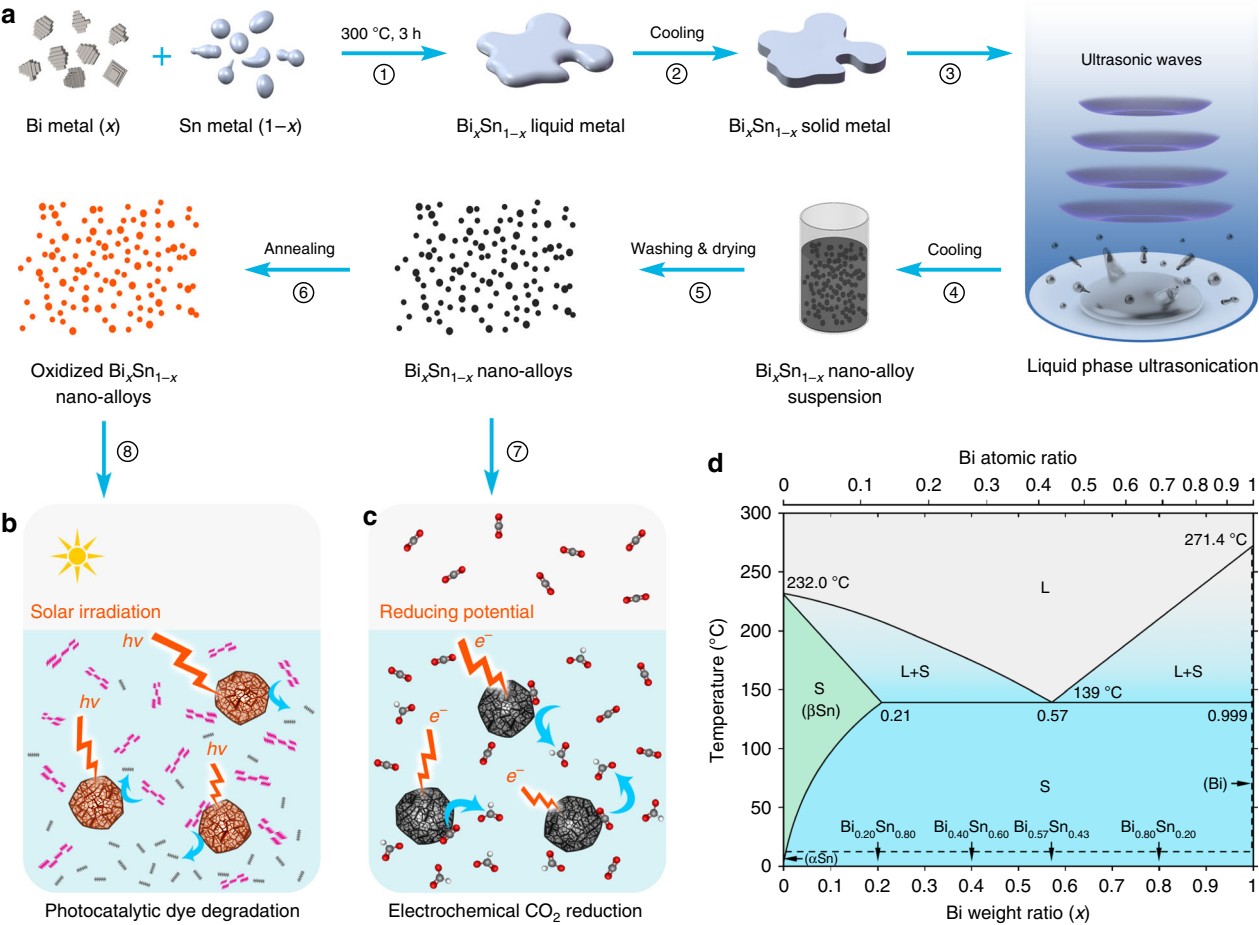

**Fig. 1** Schematics of the strategy for fabricating $Bi_xSn_{1-x}$ nano-alloy catalysts and the Bi–Sn phase diagram. **a** A flow chart showing the experimental procedures. **b**, **c** The Bi–Sn nano-alloys are used as catalysts for electrochemical reduction of $CO_2$ (**c**) and the annealed Bi–Sn nano-alloys are further used as catalysts for photocatalysis degradation of an organic model dye (**b**). **d** Bi–Sn binary alloy phase diagram (adapted from ref. [8]) with the compositions used in this study indicated. L: liquid phase; S: solid phase

and a broad shoulder at a higher temperature. The solidification temperature of the samples is lower than the melting temperature due to the supercooling effect[34].

From a thermodynamic perspective, the equilibrium at the interfacial areas between different phases in the eutectic mixture leads to their simultaneous solidification at a single temperature, but the phases in the non-eutectic mixtures solidify successively over a wide temperature range[35]. Therefore, simultaneous solidification is naturally favoured for the Bi and Sn phases in the eutectic sample to adapt more uniform distributions. Moreover, simultaneous solidification will not allow stress formed between different phase domains to dissipate effectively, resulting in more stress build-up in the eutectic sample.

The X-ray diffraction (XRD) patterns in Fig. 2e show that the two metallic phases observed in the as-prepared Bi–Sn bulk alloys are rhombohedral Bi and tetragonal Sn, which can be assigned to the crystallographic structure of elemental Bi (PDF: 04-006-7762) and elemental Sn (PDF: 04-008-4977), respectively. No inter-metallic phase is observed due to the nature of the Bi–Sn system[8]. The match between the XRD patterns of each metallic phase in the solidified alloys and their corresponding single metal XRD peaks is also an evidence of phase separation. The varying sample ratios are well reflected by the relative intensities of the Bi and Sn peaks.

In the Bi–Sn binary alloy system, the oxidation of the liquid metal surface competes between the formation of SnO, $SnO_2$, and $Bi_2O_3$. This is assumed to be governed by the Gibbs free energy of

oxidation when no sonication is applied[15]. To assess this assumption, we exfoliate the surface oxide layers formed on different liquid $Bi_xSn_{1-x}$ bulk samples (see the "Methods" section) to examine which oxide phase is selectively formed (Fig. 2f, Supplementary Fig. 2)[15–17]. The Raman spectra of the surface oxide layers (Fig. 2g) show that the result of the competition of oxide formation on the surface of the bulk Bi–Sn alloys is preferentially won by SnO. Here $Bi_2O_3$ layer is rarely seen, even at high Bi concentrations, and $SnO_2$ only reveals itself at $x = 0.80$. The favourability of SnO formation on the surface of the bulk is further validated through thermodynamic calculation using FACT-Sage[36]. As can be seen from Fig. 2h, the formation of SnO can be predicted from the Bi–Sn–O phase diagram at low $O_2$ concentration (region i) which is matched by our experimental conditions. In addition, the formation of $SnO_2$ at low Sn ratio (regions ii and iii) in the system can also be inferred from the phase diagram.

The characterisations of the bulk samples show that composition plays a significant role during the solidification, phase separation, and surface oxidation of the bulk samples, which influences their micro- and nano-structures. Importantly, mixing Bi and Sn at the eutectic ratio is found to give rise to higher structural uniformity within the bulk.

**Bi–Sn nano-alloys**. To investigate the compositional influences when the sample size is reduced to the nanoscale, the Bi–Sn nano-

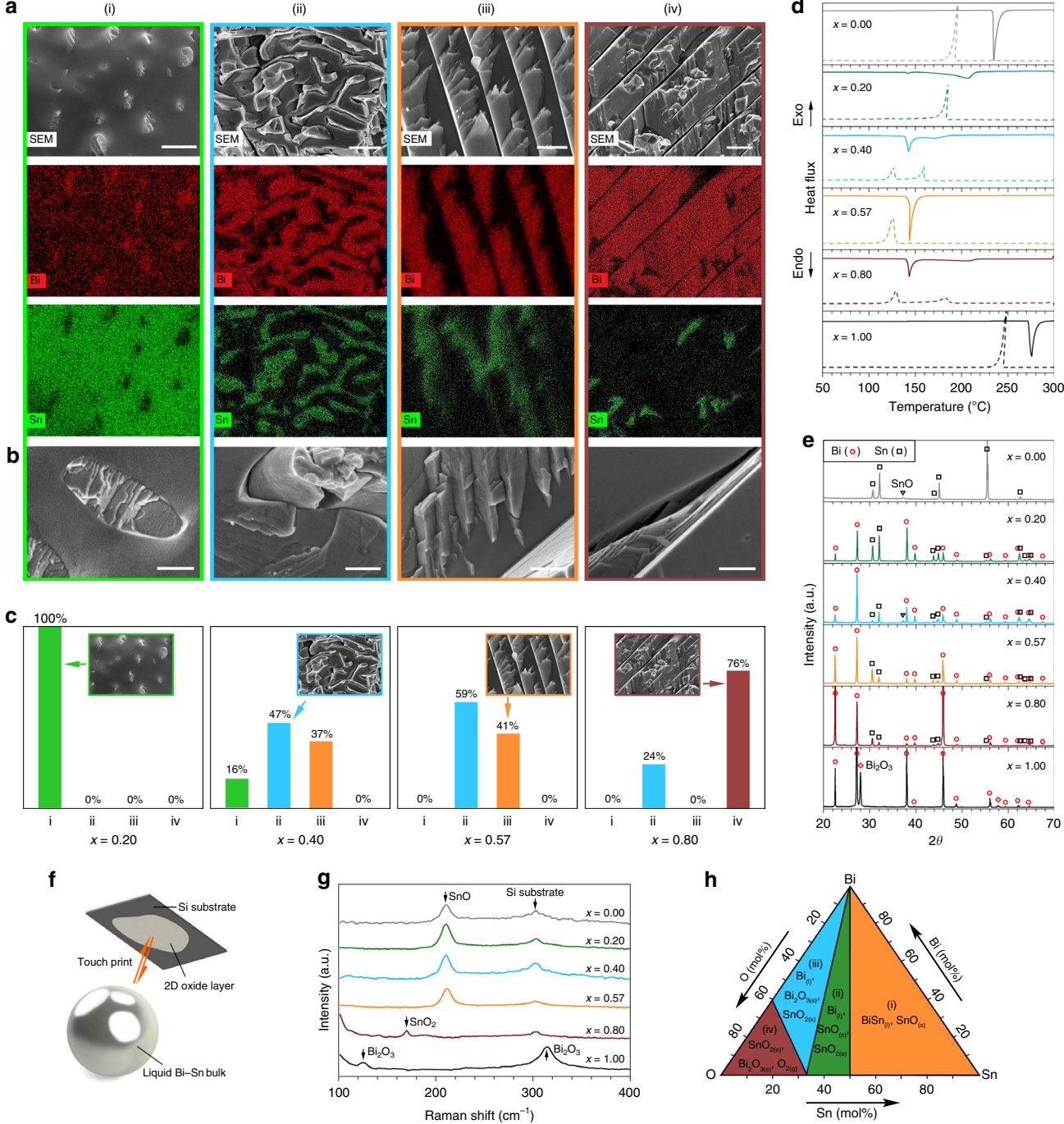

**Fig. 2** Characterisation of Bi$_x$Sn$_{1-x}$ bulk alloys. **a** SEM images and EDX element mappings showing the distribution of the Bi and Sn phases in four types of solidification structures observed in the Bi–Sn bulk alloys. Scale bars: (i) 2 μm; (ii) 5 μm; (iii) 2 μm; (iv) 5 μm. **b** Magnified views of different solidification structures. Scale bars: 500 nm. **c** Distribution of different solidification structures in different Bi–Sn bulk alloys. **d** DSC curves showing the melting (solid lines) and solidification (dash lines) trends of the bulk samples. **e** XRD patterns of the bulk samples. **f** Schematics of touch-printed surface oxide layer from liquid Bi–Sn bulk samples. **g** Raman spectra of the touch-printed surface oxide layers from bulk liquid Bi–Sn samples. **h** Phase diagram of the Bi-Sn-O system calculated at 300 °C, 1 atm

alloys prepared by liquid phase ultrasonication (Supplementary Fig. 3), are further characterised and compared. It can be seen from the transmission electron microscopy (TEM) images in Fig. 3a–d that sonicating the Bi–Sn liquid alloys at elevated temperature brings the sample size down to the nanoscale (see the "Methods" section). After post-processing, these fully solidified nano-alloy particles feature truncated spherical shapes. The size distributions show these particles range from several tens of

nanometres to about 100 nm (Fig. 3e–h). Some larger particles typically exceeding 200 nm are also occasionally found (Supplementary Fig. 4). A comparison of the particle size distribution between different samples shows a decreasing trend of the overall particle size with respect to the Bi ratio $x$. Such particle size distribution is attributed to the decrease of surface tension as the Bi ratio increases[37]. The characteristic droplet size $d$ produced by the shear flow during ultrasonication correlates with

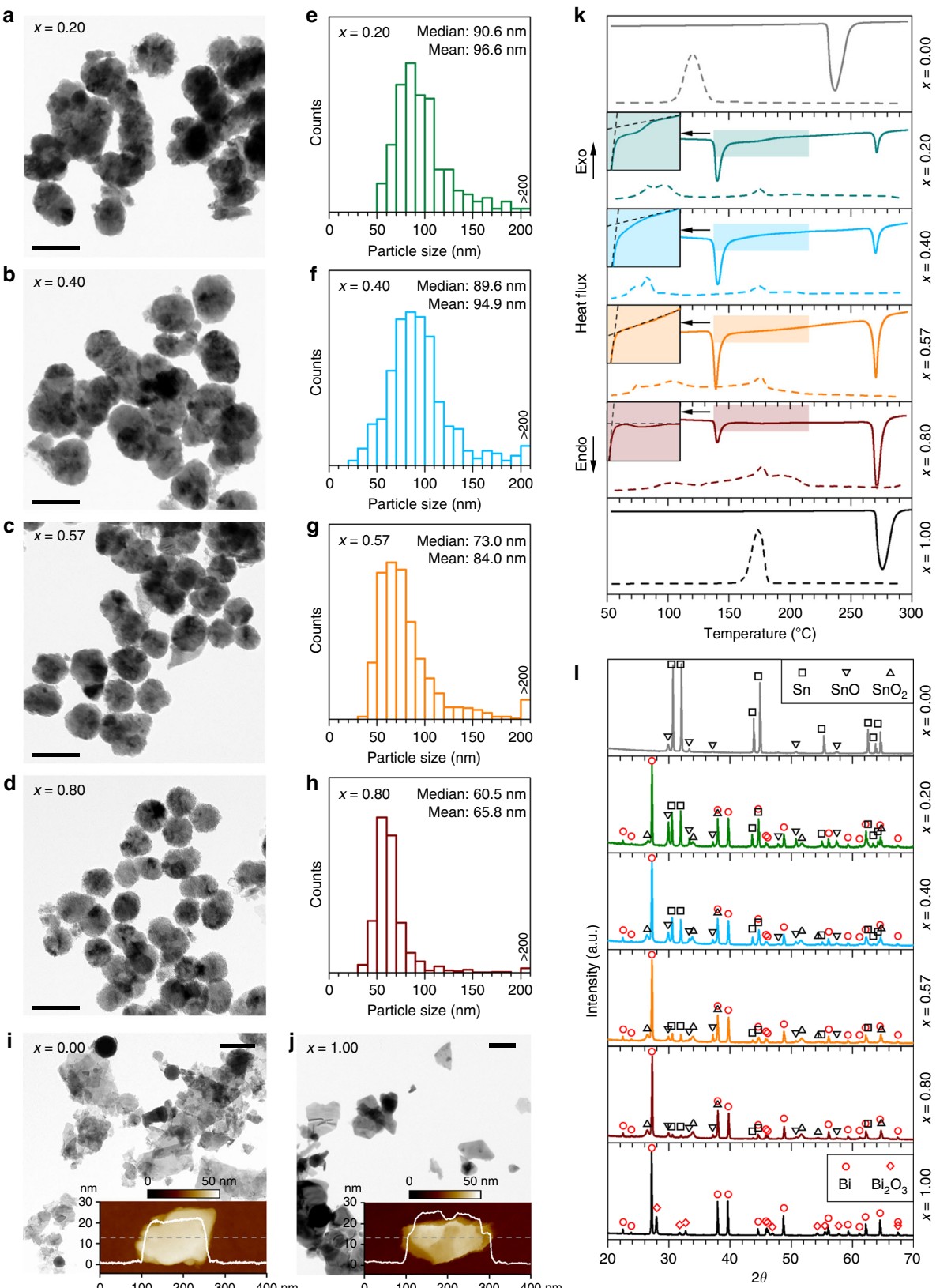

**Fig. 3** Characterisation of the Bi$_x$Sn$_{1-x}$ nano-alloys. **a–d** TEM images of the nano-alloys prepared by liquid phase ultrasonication of Bi–Sn bulk alloys with different compositions. Scale bars: 100 nm. **e–h** Size distribution of the particles with their median and mean values indicated. **i**, **j** Control samples prepared with Sn metal (**i**) and Bi metal (**j**). The inset figures show the AFM topography and the thickness profiles along the dash line of the Sn and Bi particles. Scale bars: 200 nm. **k** DSC curves showing the melting (solid lines) and solidification (dash lines) trends of the samples. **l** XRD patterns of the samples

Taylor's prediction:

$$d \propto \sigma(x)/(\eta_c \dot{\gamma}) \qquad (1)$$

where $\sigma(x)$ is taken as the Bi–Sn alloy surface tension, $\eta_c$ is the viscosity of the continuous phase (solvent), and $\dot{\gamma}$ is the shear rate of the flow produced by ultrasonication[21,38]. In our experiments, since the ultrasonication power, the solvent, and the heating conditions (temperature) are kept the same for the Bi–Sn alloys, the same solvent viscosity $\eta_c$ and shear rate $\dot{\gamma}$ can be assumed. Therefore, $d$ is expected to be only a function of the single variant $\sigma(x)$, the specimen–solvent interfacial tension. As such, the decrease of $\sigma(x)$, as the Bi ratio $x$ increases, leads to the drop in the overall particle size[37].

Interestingly, when two other control samples are prepared using either Sn or Bi metal alone, the ultrasonication process mostly produces nanoplates rather than spherical nanoparticles (spherical Sn nanoparticles are also seen occasionally, Fig. 3i). As can be seen from the atomic force microscopy (AFM) images, these nanoplates typically have a few tens of nanometre thickness, while they are a few hundred nanometres in the lateral dimensions (insets of Fig. 3i, j). This is presumably due to the formation of two-dimensional oxide of these liquid metals during the sonication process, which can be naturally delaminated from their surface. The results show one of the stark differences between processing liquid alloys and pure metals. It seems that the layering of the surface, which is seen for pure liquid metals, promotes the formation of the flakes[13,23] and the coexistence of Bi and Sn in the liquid alloys is not favourable for the metals to separate themselves from the bulk as flakes during ultrasonication.

While the DSC curves of the control samples ($x = 0.00$ and $x = 1.00$) shows similar phase transition behaviour as their bulks (but more significant supercooling), the DSC melting curves of the Bi–Sn nano-alloys split into two major peaks (Fig. 3k). This unusual observation suggests that likely two sub-particle types with different compositions co-exist in the samples. Such a conclusion is confirmed by observations in Fig. 3a–d, Supplementary Fig. 4. Peculiar phase separation occurs during sonication has also been shown by Tang et al.[39]. These results imply that when the liquid bulk alloys are fragmented into nanodroplets during ultrasonication, the localised composition may be changed. The high-melting-temperature phase (in correspondence to the DSC peak on the right) is associated with the large particles with Bi-rich cores, which have different Bi–Sn ratio after the separation from the bulk (Supplementary Fig. 5). The other low-melting-temperature phase (in correspondence to the DSC peak on the left) is assigned to particles that are smaller in size (Supplementary Fig. 6). It is also confirmed by EDX (Supplementary Fig. 7) that the phase with the melting onset at ~265 °C, 6 °C lower than that of bulk Bi, is a Bi-rich phase rather than elemental Bi. These results mean that despite the compositional changes, all particles either small or large, constitute both Bi and Sn. Since the small particles, which account for the majority of the surface of the samples, play the leading role in chemical or electrochemical reactions, we proceed with a focus on the analysis of the small particles.

It can be seen from the melting peaks of the small particles (the DSC peak on the left in Fig. 3k) that, in comparison with the DSC curves of the bulk samples (Fig. 2d), the shoulders of the non-eutectic samples are shaved after the liquid phase ultrasonication, indicating the accompanying compositional changes. By contrast, as shown in the insets of Fig. 3k (see also Supplementary Fig. 8 for comparison of the temperature derivative of heat flow), the single melting peak of the eutectic sample implies that the eutectic Bi–Sn ratio in the small particles is still maintained after ultrasonication. The result in fact shows how differently the eutectic sample behaves in comparison to the non-eutectics as it

seems that finer nucleation occurs during eutectic solidification, dominating the behaviour of the phase separation. During the cooling half cycle, the nucleation process is sensitive to multiple factors such as the size, existence of surface oxides, and cooling conditions for small particles[40], and as such the Bi–Sn nano-alloy samples show complex solidification behaviour.

The XRD technique, which has deep penetration, allows a glimpse of the inner particle compositions of the particle samples. More importantly, the outcome will tell whether the knowledge from the XRD results of the bulk alloy can be applied to the core of nano-alloys. Encouragingly, as shown in Fig. 3l, the XRD patterns reveal that the Bi and Sn metallic phases in the particles share the same crystallographic phases with their bulk precursors and the samples are partially oxidised after ultrasonication. SnO and $Bi_2O_3$ are recognised as the oxide phases for the control Sn and Bi samples, respectively. For the Bi–Sn nano-alloys, two types of Sn oxides, namely SnO and $SnO_2$, emerge, but the oxide of Bi is absent. The absence of Bi oxide deep inside the nano-alloys can be rationalised by considering the competing oxidising process of the metallic phases based on the Bi–Sn–O phase diagram (Fig. 2h). The oxidation mechanisms of pure Sn and Bi is likely due to the constant exfoliation of the flakes during the sonication, which is also similarly seen in the sonication of Ga[23]. The difference in morphologies between the pure and alloyed samples shows the significant impact of alloying, which increases the surface entropy and disturbs its order, reducing the possibility of removing surface layers by the shear force during the ultrasonication process.

The surface of nanoparticles may constitute different compositions in comparison to their core. Understanding the surface composition is especially important for nano-alloys as different oxides can emerge on or within nanoparticles, while they are not seen in the bulk. The coexistence of Bi and Sn oxides on the surface of the Bi–Sn nano-alloy samples is confirmed further by EDX-coupled scanning TEM (STEM) and X-ray photoelectron spectroscopy (XPS), both of which are surface sensitive techniques. STEM–EDX mapping shows the distribution of Bi and Sn, together with O due to partial oxidation (Fig. 4a, Supplementary Figs. 5–7). The XPS twin peaks at the Bi 4$f$ region (159.4 and 164.6 eV) and the Sn 3$d$ region (486.7 and 495.2 eV) are characteristic features of $Bi_2O_3$, and $SnO/SnO_2$, respectively (Fig. 4b)[41,42]. From the variation of the intensity of the Bi 4$f$ and Sn 3$d$ peaks, it can be inferred that the alloys' compositional influences are peculiarly expressed on the surface of the nano-alloys. The significant difference of the relative intensity of O1$s$ peaks between the nano-alloy samples and the control Bi and Sn samples may come from the different oxygen dissolvability of the solvents that used to prepare the samples (see the "Methods" section).

Given the overlapping of the Sn 3$d$ XPS peaks of SnO and $SnO_2$, Raman spectroscopy was further engaged to access the composition of the Bi–Sn nano-alloys (Fig. 4c). The two major Raman peaks at 113 and 211 cm$^{-1}$, which are the characteristic $E_g$ and $A_{1g}$ modes of SnO, respectively, indicate that the surface of the nano-alloys is dominated by SnO[17,43]. Interestingly, with increasing Bi ratio (and simultaneously decreasing Sn ratio), small peaks around 160 and 190 cm$^{-1}$ begin to emerge for the eutectic sample, which can be assigned to $SnO_2$[44] and $Bi_2O_3$[45], respectively. The emergence of $SnO_2$ and $Bi_2O_3$ peaks can be expected given that the decrease of the Sn content in the Bi–Sn nano-alloys enhances the propensity of further oxidisation of SnO to $SnO_2$ and increase of Bi augments the possibility of $Bi_2O_3$ emergence. It thus can be inferred that the $SnO_2$ content, as revealed by XRD (Fig. 3l), mainly belongs to the region under the nano-alloy surface.

Altogether, the nano-alloys prepared by liquid phase ultrasonication feature a $Bi_2O_3$-doped SnO surface with their doping

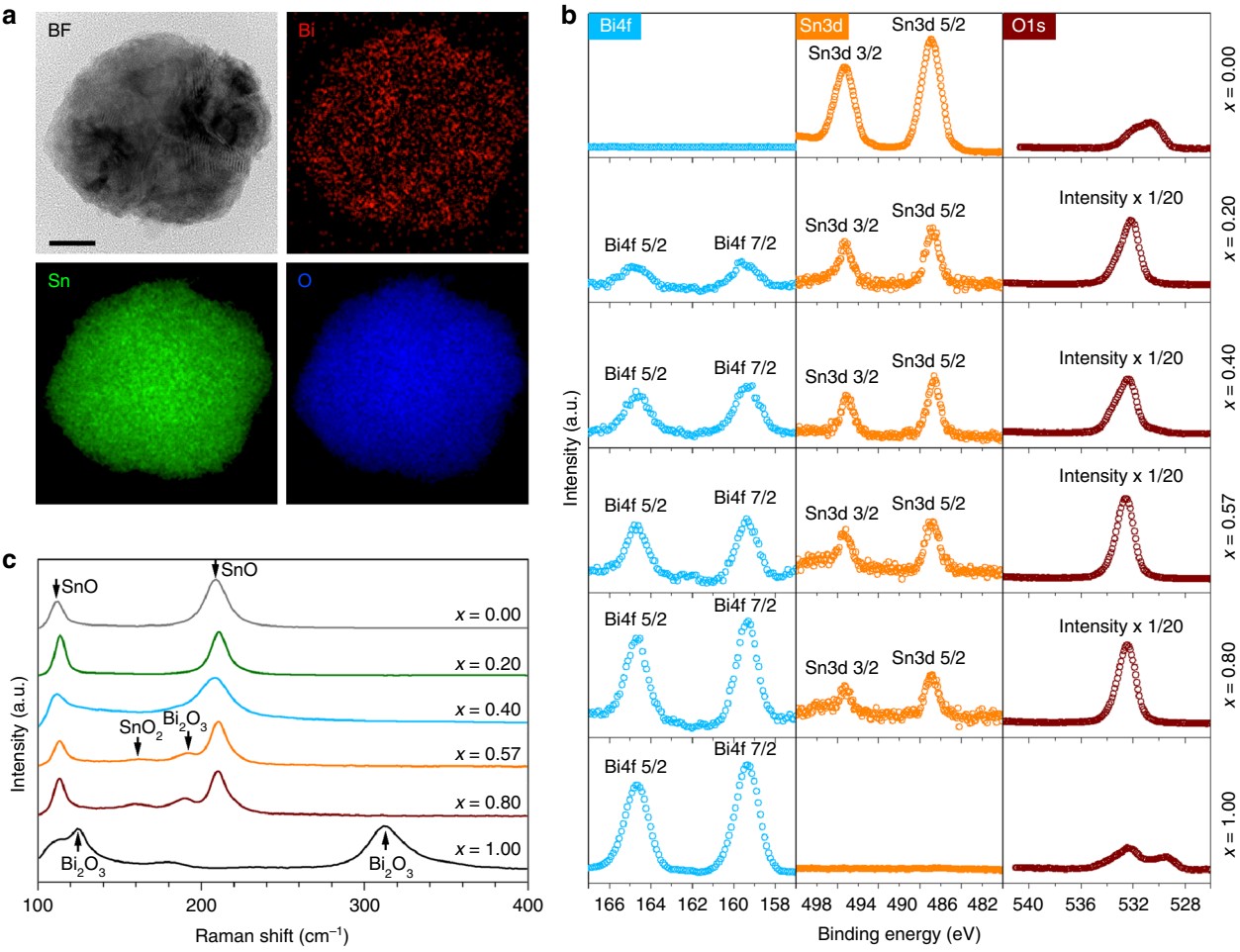

**Fig. 4** Surface composition analysis of the $Bi_xSn_{1-x}$ nano-alloys. **a** TEM image and STEM–EDX element mapping, showing the distribution of Bi, Sn and O in a eutectic nano-alloy particle. Scale bar: 20 nm. **b** XPS spectra of different Bi–Sn nano-alloys. The same scale is used for the XPS intensity for different elements of the same sample unless otherwise specified in the figures. **c** Raman spectra of different Bi–Sn nano-alloys

ratio proportional to the Bi concentration (according to XPS). The influence of the Bi–Sn ratio on the surface composition again reveals that the eutectic ratio is likely the critical turning point in which the emergence of both $SnO_2$ and $Bi_2O_3$ becomes more effective (according to Raman spectroscopy). The formation of such peculiar structures on the surface of the nano-alloys can be described by the selective surface migration, phase separation and oxidation of the nano-alloy constitutes during liquid phase ultrasonication[15]. One important observation from these experiments is the difference between the surface oxidation of the bulk and oxides on nano-alloy surface. This deviation is presumably caused by the sonication process, which promotes phase separation at the same time as the surface oxidation takes place. We have already discussed about the surface domination of the bulk sample by SnO when no sonication is applied. However, our characterisations revealed that under ultrasonication $SnO_2$ also appears near the surface when Sn concentration reaches the eutectic value or lower. For these concentrations the emergence of $SnO_2$ is favoured, especially when oxygen can be continuously dissolved in the solvent, which is the case for the sonication process.

The mixing of different atom species in alloys can offer more intrinsic influences at the atomic scale. The dark-field TEM (DF-TEM) images in Fig. 5a–d show that the $Bi_xSn_{1-x}$ nano-alloys produced by our method are polycrystalline. Polycrystallinity implies the presence of grains and grain boundaries[46]. Qualitative comparisons between the grain dimensions within the samples

reveal that the phase distribution in the eutectic nanoparticles are more homogeneous than the non-eutectics at the equivalent scale and as such constitutes smaller grains (more DF-TEM results can be found in Supplementary Fig. 9). These results mean that, after solidification, phase separation imposes itself more delicately on the eutectic nano-alloys. As shown in Fig. 5e–h, we further outline the grain boundaries of the bright-field high-resolution TEM (BF-HR-TEM) images taken from different nano-alloy samples according to different orientations of crystal lattices of different grains. Then the size of the grains within individual nanoparticles can be statistically evaluated. The surveys of the grain size distribution of different samples reveal that the overall grain size of the eutectic sample is the smallest among all the samples (insets of Fig. 5e–h), while its particle size is not (Fig. 3e–h). Crystal grains imply two-dimensional defects. Logically, the eutectic sample with the smallest grains should have the largest amount of boundaries and interfaces. We note that the higher polycrystallinity (smaller grain size) in the eutectic nano-alloys is typically found in the HR-TEM sample images (Supplementary Fig. 10).

In addition, the formation of low-dimensional defects is frequently observed in the eutectic nano-alloys, which is either absent or rarely seen in the non-eutectic particles. For instance, classic one-dimensional edge dislocations are observed (Fig. 5i, j) at much higher frequencies of occurrence in the eutectic nano-nanoalloys. It can also be seen from Fig. 5i that due to the small scale of grains and the coexistence of local edge dislocations, their

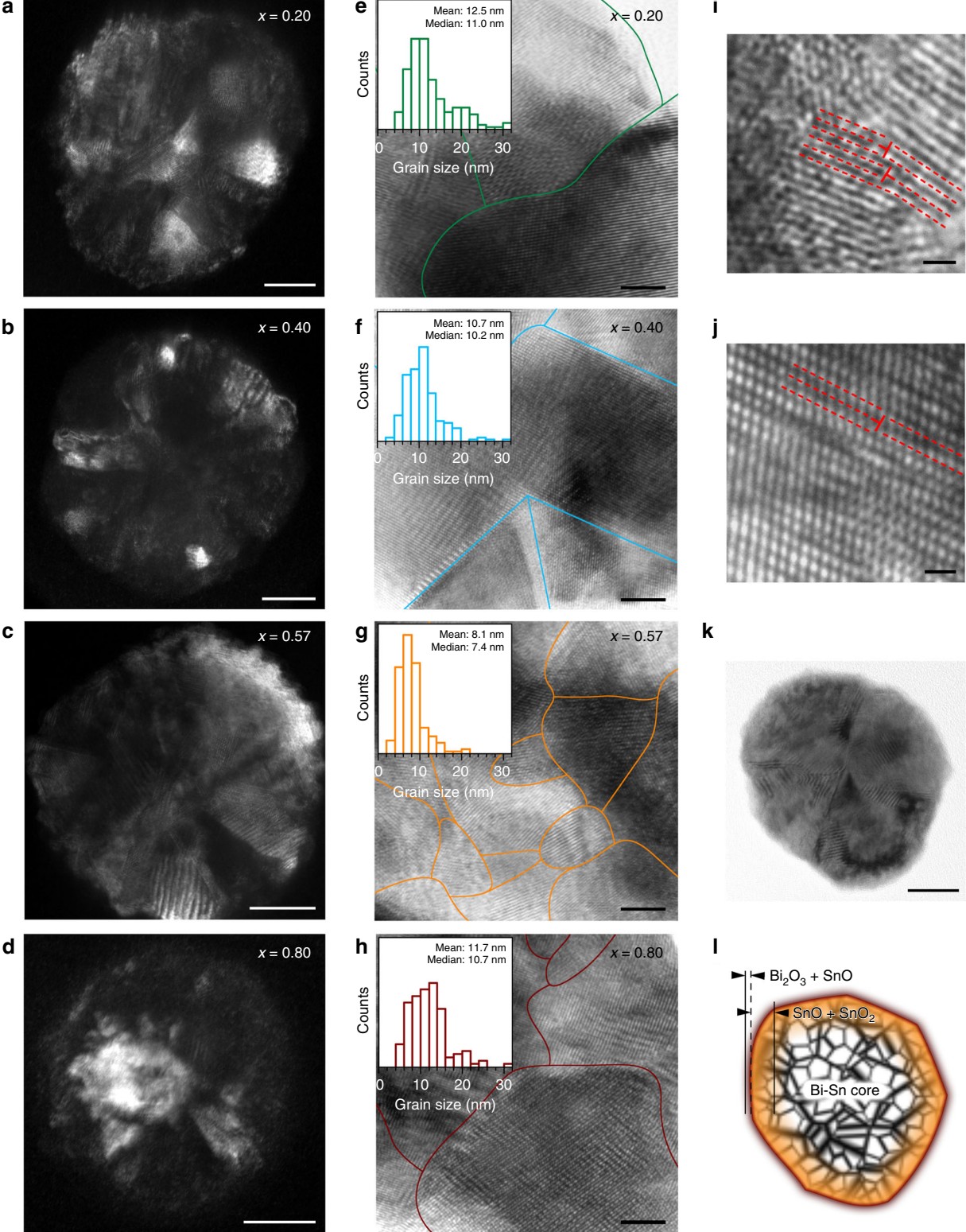

**Fig. 5** Crystallographic characterisation of the Bi$_x$Sn$_{1-x}$ nano-alloys. **a–d** DF-TEM images of individual particle of different Bi–Sn nano-alloy samples. Scale bars: 20 nm. **e–h** BF-HR-TEM images of the nano-alloys with their grain boundaries outlined. The insets present their respective grain size distribution. Scale bars: 5 nm. **i, j** BF-HR-TEM images showing line defects (edge dislocations) indicated by T-shape symbols that observed in the eutectic nano-alloys. Scale bars: 1 nm. **k, l** TEM image of a eutectic nano-alloy particle (**k**), scale bar: 20 nm, and an illustration shows its heterostructures based on the characterisation results (**l**)

glide planes terminate shortly. Figure 5j shows a more complex scenario in which long-range lattice misalignments occur on both sides of the indicated edge dislocation. Presumably, this is formed as screw dislocations set in and couple with the edge dislocations (mixed dislocations)[47], or by faulted stacking of different atom species. Based on all the characterisations of the nano-alloys, an illustration of the characteristic structure of the nano-alloys is presented in Fig. 5k, l. The nano-alloys have a surface containing SnO and $Bi_2O_3$, a deeper region mostly composed of SnO and $SnO_2$, and a Bi–Sn metallic core. All these regions feature intense defects.

Defect formation is known to be facilitated in multi-metal systems due to differences in radius and electron structure of the atoms[48]. The Bi–Sn nano-alloys prepared by liquid phase sonication are expected to have enhanced defect formation since, besides the incorporation of different atom species, their crystal structures are established through a phase transition process[35]. The distinct crystallisation behaviours between the eutectic and non-eutectic samples are thought to be responsible for their structural differences after solidification. The solidification of the Bi–Sn alloys is governed by the formation of nuclei and their subsequent growth, with the later process being much faster than the former[32]. Therefore, the solidification of the hypoeutectic ($x = 0.20$ and $0.40$) and the hypereutectic ($x = 0.80$) Bi–Sn alloys start with the nucleation and growth of Sn and Bi, respectively. Since the growth of crystals takes place preferentially on crystalline faces and the Bi–Sn system forms no intermetallic phase, it results in a gradual growth of Sn or Bi during the solidification, which leads to the formation of larger grains hence lower grain density for the non-eutectic alloys. For the eutectic ($x = 0.57$) alloy, the two types of nuclei form simultaneously during solidification, and the successive growth of these nuclei progresses rapidly. As a result, the separation of the Bi and Sn phases takes place more locally, leading to smaller grain sizes and higher intensity of grain boundaries than the non-eutectics.

The above-mentioned thermodynamics of alloy solidification applies to both the Bi–Sn nano-alloys and bulk alloys, so some observations during the solidification of Bi–Sn bulk alloys can be associated to that of the nano-alloys. As can be seen from the zoom-in features of the bulk samples after solidification in Fig. 2b, significant stress build-up in the bulk eutectic sample is established from the highly distorted fibrous structures. This is in agreement with more enhanced defect formation in the eutectic nano-alloys, resulting in smaller grains in general (Fig. 5e–h). For the non-eutectic samples, the gradual growth during solidification allows the establishment of large rods or lamellae structures in bulk samples, that are relatively stress free at their grain boundaries.

Similarly, for nano-alloys, this is associated with the formation of relatively larger grains of Bi and Sn.

Different phases in the bulk solid samples features micrometre separated regions (Fig. 2a, b), while those of the nano-alloys are typically around ten nanometres (Fig. 5e–h). However, considering the differences between the nano-alloy and bulk alloy solidification, it is likely that the heterogeneity of the nano-alloys further results from: (1) the extra energy applied during the sonication, and (2) possibility of liquid and solid phase coexistence in small dimensions that has been reported previously[49]. A such, the nano-alloys constitute much more finely mixed grains.

**Electrochemical activity of the Bi–Sn nano-alloys.** Since crystallographic defects can act as active sites during chemical reactions and improve the performance of catalysts[50–52], higher polycrystallinity and smaller grains should also lead to better catalytic performance. The atomic-scale defect containing eutectic Bi–Sn nano-alloy, with the smallest grains and the highest frequency of line defects, is therefore expected to be more catalytically active in comparison to its non-eutectic counterparts.

To validate this assumption, we first investigate the $CO_2$ reduction reaction ($CO_2RR$) to test the activity and selectivity of the eutectic nano-alloys towards formate ($HCOO^-$) formation[41,42,53]. The results of the potentiostatic experiments carried out with different Bi–Sn nano-alloy samples are presented in Fig. 6. The polarisation curves obtained with the eutectic nano-alloy (Fig. 6a) reveals an enhanced current density ($j$) under $CO_2$, in comparison with Ar saturated 0.1 M $KHCO_3$, attaining a value of $-33$ mA cm$^{-2}$ at an applied potential of $-1.4$ V vs. reversible hydrogen electrode (RHE). The inset of Fig. 6a shows a representative chronoamperometric $j$–$t$ curve obtained at $-1.2$ V which indicates steady $j$ during the bulk electrolysis sessions. Two gas phases $H_2$ and CO are detected using gas chromatography. Nuclear magnetic resonance (NMR) test reveals that the only liquid product generated during $CO_2RR$ is the target formate. From Fig. 6b, it is evident that the eutectic nano-alloy demonstrates a strong potential dependence for the Faradaic efficiency for formate ($FE_{HCOO^-}$) and that formate can be detected at an applied potential of $-0.9$ V with a Faradic efficiency of 38%. Further increasing the applied potential leads to an increase in $FE_{HCOO^-}$, attaining a maximum of 78% at $-1.1$ V before declining to ~60% at $-1.2$ V. When compared with the control Sn and Bi samples (Supplementary Fig. 11), the eutectic nanoalloy is found to be more selective towards $HCOO^-$ generation and its overall activity is comparable to the benchmark

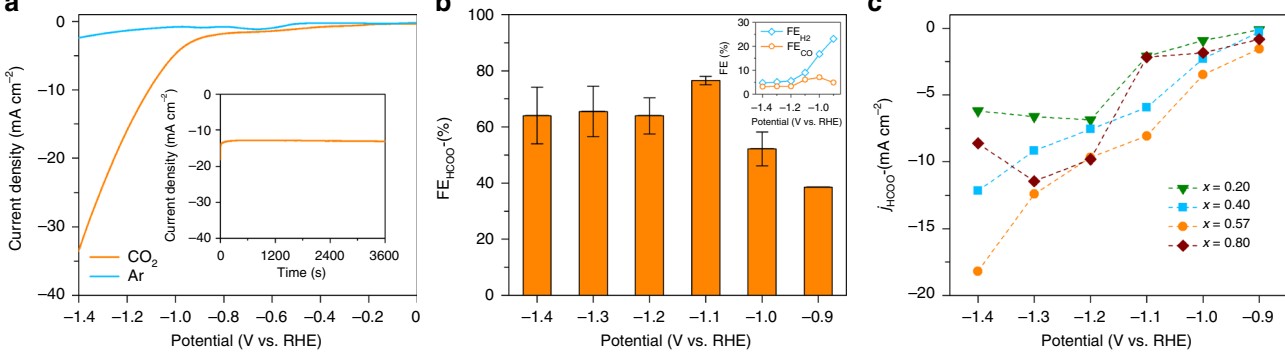

**Fig. 6** $CO_2RR$ activity of the Bi$_x$Sn$_{1-x}$ nano-alloy catalysts. **a** Linear sweep voltammetry (scan rate: 5 mV s$^{-1}$) of the eutectic sample in $CO_2$ and Ar saturated 0.1 M $KHCO_3$. The inset shows a chronoamperometric $j$–$t$ curve at an applied potential of $-1.2$ V vs. RHE. **b** Dependence of $FE_{HCOO^-}$ with applied potential for the eutectic sample in $CO_2$ saturated 0.1 M $KHCO_3$. The inset shows the dependence of $FE_{H2}$ and $FE_{CO}$ of the eutectic nano-alloy over the same potential range. **c** Comparison of $j_{HCOO^-}$ of different Bi–Sn nano-alloy samples as a function of the applied potential in $CO_2$ saturated 0.1 M $KHCO_3$

Sn and Bi catalysts in the literature (Supplementary Table S1). Moreover, the eutectic nano-alloy is capable of suppressing the competing hydrogen evolution reaction (HER) that typically accompanies $CO_2RR$[54], as indicated by the low Faradaic efficiency for $H_2$ ($FE_{H2}$) across the potentials tested herein (inset of Fig. 6b). The maximum $FE_{H2}$ attained with eutectic nano-alloy was 23% at $-0.9$ V which decreases steadily to <10% as the applied potential is lowered to $-1.1$ V. Furthermore, the Faradaic efficiency for CO ($FE_{CO}$) remains at <7% throughout all the potentials tested. The summations of Faradaic efficiencies of the products (gas products alongside liquid products) do not add up to 100%, especially for applied potentials beyond the optimum value ($-1.1$ V) and this loss of Faradaic efficiency is ascribed to the charge loss due to the formation of slight carbonaceous species[55], as identified by post-reaction SEM and Raman measurements (Supplementary Fig. 12). These tests show the strong viability of the eutectic nano-alloy sample for $CO_2RR$.

We then carry out a systematic investigation to compare all the Bi–Sn nano-alloy samples for $CO_2RR$. These collective electrocatalytic results indicate that the Bi–Sn nano-alloy catalysts generally demonstrate higher selectivity towards $CO_2RR$ than HER (Fig. 6c, Supplementary Fig. 13). Remarkably, the eutectic sample always gives higher partial current densities towards the target product of $HCOO^-$ ($j_{HCOO^-}$) at all applied potentials. Since the size distributions (which is the smallest for $x = 0.80$ and not the eutectic) rule out the possibility of size contribution to the superior performance, the structural differences thus should be primarily responsible for the eutectic to enable higher activity towards $CO_2RR$. The pronounced defects formation observed in the eutectic nano-alloy sample is the most conceivable underlying mechanism that can be envisaged.

The ability of an electrocatalyst to produce a specific product during $CO_2RR$ can be explained by the stabilisation of different reaction intermediates on a catalyst surface. In the case of Bi and Sn catalysts, $HCOO^-$ is preferentially formed during $CO_2RR$[56]. $CO_2RR$ on Bi and Sn catalysts proceed via the first electron transfer to the $CO_2$ reactant to form $CO_2$ anion radical intermediate ($^*CO_2^{\cdot-}$), which is generally accepted to be the rate-determining step for $CO_2RR$[56]. Subsequent proton and electron transfers lead to the formation of $HCOO^-$ through either bidentate intermediate ($^*OCHO$) or adsorbed carboxyl ($^*COOH$) species[57,58]. For the Bi–Sn nano-alloys, both the synergistic effect between Bi and Sn on the surface electric state and the crystallographic defects of the catalysts can affect $CO_2RR$. On one hand, the presence of a metal (e.g. Bi) that is more electronegative than the parent metal itself (e.g. Sn) improves the selectivity of the catalyst towards formate, since the $p$ and $d$ orbitals of Sn electron states can be upshifted away from the Fermi level, leading to faster electron transport to the $CO_2$ reactant[41,59,60]. On the other hand, defects such as grain boundaries can act as active sites for $CO_2RR$ as a result of their favourable electrical and chemical properties[52]. These grain boundaries are reported to tune the binding energy of the $CO_2RR$ reaction intermediates, thereby increasing the formate current density $j_{HCOO^-}$[61]. Therefore, the observed enhanced $j_{HCOO^-}$ of the eutectic nano-alloy sample (Fig. 6c) suggests that defects play a major role here, which matches the increased grain boundary intensity within the eutectic sample.

**Photocatalytic activity of the Bi–Sn nano-alloys.** Given that both Bi oxide and Sn oxide are photocatalytically active[62,63], we conduct one further annealing step to convert the Bi–Sn nano-alloys to their oxide phases for dye degradation, so as to demonstrate the multifunctionality of the materials produced by liquid phase ultrasonication. The as-prepared dark grey Bi–Sn

nano-alloys became yellowish after annealing and XRD measurement shows that the annealing process fully converts the sample to $Bi_2O_3$ and $SnO_2$ (Fig. 7a). Additionally, a binary oxide phase $Bi_2Sn_2O_7$, which is generally recognised as a visible-light-driven photocatalyst[64], is also generated, indicating that our method can access more complex catalytic structures. As shown in Supplementary Fig. 14, more $Bi_2Sn_2O_7$ is detected in the annealed eutectic sample. Most likely, the scenario that describes this increase is that after annealing, a large amount of the binary oxide phase of $Bi_2Sn_2O_7$ is established at the intense grain boundaries of the Bi and Sn phases in the eutectic sample, where the two initially metallic phases are in contact with each other.

As shown in Fig. 7b, when compared to the Raman spectra of the control Sn and Bi samples, the predominance of $Bi_2O_3$[45,65] and $SnO_2$[44,66] in the annealed nano-alloy samples is further confirmed. We highlight that the Raman results reveal that the annealed eutectic sample has the highest relative intensity of $SnO_2$ defects characterised by $I_D/I_A$, where $I_D$ (577 cm$^{-1}$) and $I_A$ (633 cm$^{-1}$) is the $SnO_2$ defect and active Raman mode, respectively[67]. The results provide another evidence to support our crystallographic defect characterisations that the eutectic nano-alloy sample has more enhanced defects formation than the non-eutectics.

As shown in Fig. 7c, d, the plots of ln ($c_0/c_t$) vs. $t$ indicate that the annealed nano-alloys demonstrate a pseudo-first-order degradation kinetics towards the Congo red model dye, following the relation:

$$-\frac{dc}{dt} = kc \qquad (2)$$

where $c$ is the concentration of dye and $k$ the observed rate constant (slope of the ln ($c_0/c_t$) $- t$ curves). As shown in Fig. 7c, tests on different annealing temperatures reveal that the eutectic sample offers the highest degradation when annealed at 500 °C, and this annealing temperature is chosen for other samples. Comparison between different sample ratios again shows that the annealed eutectic sample has the highest rate constant $k$, indicating its highest dye degradation rate (Fig. 7d). In addition, the nano-alloy samples demonstrate higher activity than the control Sn ($x = 0.00$) and Bi ($x = 1.00$) samples (Supplementary Fig. 15). The good correlation between the defect intensity ratio ($I_D/I_A$) and the rate constant $k$ (Fig. 7e) variation as a function of $x$, and that both values reach maxima for the eutectic sample, indicate that the Sn-related defects may also play a role in enhancing the number of active sites for the photocatalysis. This enhanced number of active sites, together with the maximum intensity of the binary compound $Bi_2Sn_2O_7$ (Supplementary Fig. 14), can be attributed to the augmented photocatalytic effect for the annealed eutectic sample.

The characterisation and the photocatalytic experiment of the annealed samples show that our strategy of processing alloys in their liquid state can produce nano-alloy (partially oxidised) electrochemical catalysts, and be further utilised to create oxide photocatalytic catalysts. Also, the Raman results provide a spectroscopic proof to support our TEM microscopic defect characterisations that processing the eutectic composition leads to more intense defect formation. Therefore, our results suggest that it may be a general trend for the eutectic ratio to be the optimum for developing catalysts using liquid phase ultrasonication. The phase transition behaviour of the eutectic alloy, which has shown to be responsible for its unique solidification structures across different length scales and the formation of different types of defects, even after annealing, leading to its superior performance. Since the eutectic sample also has the lowest possible melting temperature, the findings provide an appealing evidence that favours eutectic systems.

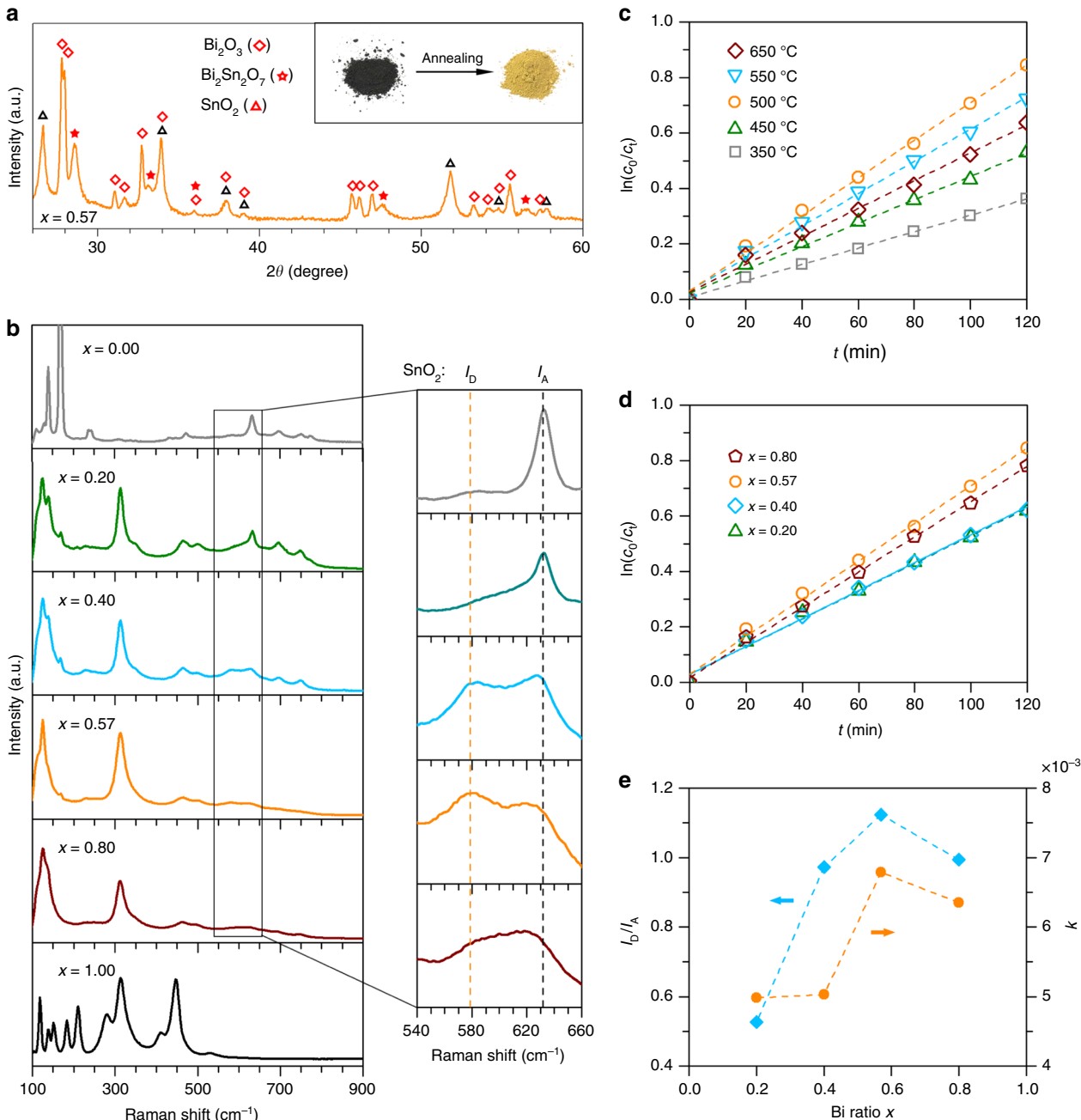

**Fig. 7** Characterisation and photocatalytic activity of the annealed $Bi_xSn_{1-x}$ nano-alloys. **a** XRD patterns of the eutectic sample after annealing at 500 °C for 1 h. The inset shows the colour of the eutectic sample before and after annealing. **b** Raman spectra of the Bi–Sn nano-alloys annealed at 500 °C for 1 h. The magnified regions show the relative intensity of the $SnO_2$ defect mode $I_D$ and active mode $I_A$. **c** Plots of ln $(c_0/c_t)$ vs. $t$ for the eutectic samples annealed at different temperatures. **d** Plots of ln $(c_0/c_t)$ vs. $t$ for the Bi–Sn nano-alloy samples annealed at different 500 °C for 1 h. **e** The dependence of $SnO_2$ defect intensity (characterised by $I_D/I_A$) and dye degradation rate constant $k$ on the mixing ratio of the Bi–Sn nano-alloy samples

## Discussion

Using the Bi–Sn alloy system as a model, we have demonstrated some of the fundamental principles governing nanotechnology-enabled alloys in a facile synthesis process. These alloys offer lower melting temperatures than their constituent metals and as such can be processed in their liquid form using relatively low energy. Here we applied ultrasonication at elevated temperature to break up the bulk of these mixtures into nano-entities for developing polycrystalline nano-alloy catalysts. This top-down nano-alloy synthesis approach shows how this binary alloy system, with no intermetallic phases, could be solidified into nano-particles with separated phases.

We studied both bulk and nano-alloys of Bi–Sn mixtures at different ratios focusing on phase separation, surface oxidation and defect formation. Different Bi–Sn ratios were found to have distinct structural influences. We showed that ultrasonication changes the type of oxides emerge on the nano-alloy surface in comparison to that of the bulk. Specifically, in the eutectic case, SnO, $SnO_2$, and $Bi_2O_3$ can all appear on or near the surface of the nano-alloys in contrast to the bulk cases where only SnO is seen. The eutectic composition also resulted in the smallest grains, as well as other types of defects within the synthesised nano-alloys. We provided evidence indicating that the eutectic ratio is the optimum mixing ratio for catalysis applications of nano-alloys, in

two examples, one for the electrochemical $CO_2RR$ and the other for the photocatalytic degradation of a model dye.

Although low-melting-point metals, including zinc group and post transition metals, seem to be the first beneficiary of the demonstrated procedure, the concept can also be extended to many other systems. This includes selected high-melting-point metals that can be incorporated in low-melting-point ones to produce alloys with significantly lower melting temperatures (see Supplementary Table 2 for a list of binary systems and Supplementary Table 3 for proposed solvents). Additionally, when incorporated, different metals will compete to attain the dominance on the surface of the alloys. This means that if chosen correctly, fractional amount of a metal in alloys can still play a significant role in chemical or catalytic reactions which mostly rely on surface properties.

The finding that eutectic nano-alloys, made from specific metals, are better catalysts in comparison to other mixing ratios of the same metals is appealing. This is due to the fact that processing liquid eutectic alloys, which possess the lowest possible melting point, is favoured for liquid metal ultrasonication and many other liquid-metal-based methods. Our investigations could help in devising strategies for large scale production of nano-alloys for various applications. In addition, the work may provide useful insights for selecting the optimum mixing ratios to enhance catalytic performance in other nanotechnology-enabled metallurgy approaches. The incorporation of a liquid phase in their processing history brings intrinsic differences in comparison to other procedures for obtaining nanoparticles. The diversity of metal species and also their combinations (binary, ternary, quaternary, etc.) forecast promising future possibilities.

## Methods

**Bi–Sn bulk alloy preparation**. The Bi–Sn weight ratios used for preparing the $Bi_xSn_{1-x}$ bulk alloy samples were $x = 0.00$, 0.20, 0.40, 0.57, 0.80, and 1.00, where $x$ denotes the Bi weight ratio. During preparation, metallic bismuth (purity > 99.99%, Rotometals, USA) and tin (purity > 99.99%, Rotometals, USA) were put in glass containers and melted in a tube furnace (TF55030C-1, Thermo Scientific). The samples were heated from room temperature to 300 °C, exceeding the melting point of both Bi (271.4 °C) and Sn (232.0 °C), and were kept at 300 °C for 3 h. During the first 1 h, the melts were shaken several times to facilitate mixing. The samples were then taken out of the furnace and cooled down in ambient air (room temperature). Under such condition the solidification of the samples was completed within a few seconds.

**Bi–Sn nano-alloy preparation**. In order to melt the bulk Bi–Sn samples while applying ultrasonic irradiation, the samples (typically 1.0 g), together with 70 mL of DMSO (Sigma-Aldrich, purity: 99.99%, boiling point: 189 °C), were heated inside a glass vial on a hotplate set to 400 °C. To prepare the Bi and Sn samples for control experiments, silicone oil (silicone 200 fluid, boiling point > 300 °C, Chem-Supply, Australia) was used. The temperature of the solvent at the sample vial bottom was measured to be 187.6 ± 0.6 °C for DMSO and 284.4 ± 0.3 °C for silicone oil, respectively. Such configurations, which caused DMSO to boil slightly (DMSO vapour is flammable, good ventilation is required, e.g. work under fumehood), enabled the melting of all the samples for ultrasonication (Supplementary Fig. 3). A probe sonicator (VC 750, Sonics & Materials) coupled with a 1/2″ diameter tip was used, and the amplitude was kept constant at 60%. The sonication was paused for 10 s after every 10 s ultrasonic irradiation during every 2-h sonication session to consistently sediment large particles. Thereafter, the particle-containing solvent was cooled down to room temperature inside a bath sonicator (FXP10M, Uni-Sonics) to match the rapid cooling condition of the bulk samples. To wash off DMSO (silicone oil for control samples) from the particles, solvent replacement was then performed repeatedly (more than five times) with deionized water (chloroform and water, successively, for control samples) in a centrifuge (CR4000, Thermoline Scientific) at 5000 rpm for 10 min. Another low-speed centrifugation at 500 rpm for 1 min was further performed to filter large particles. The final Bi–Sn nano-alloy samples were obtained by drying the supernatant at 80 °C.

**Sample characterisations**. The XRD patterns of both bulk and particle samples were obtained using an X'pert Multipurpose XRD (MPD) System ($\lambda = 1.5418$ Å, Cu-K$\alpha$ radiation). A Thermo Scientific Al K-$\alpha$ monochromated X-ray spectrometer with photon energy of 1486.7 eV and pass energy of 50 eV was used for the XPS measurement of the Bi–Sn nano-alloy samples. The DSC measurements were

carried out on a Q20 DSC (TA Instruments). A 25–300 °C temperature range was used for the all the DSC measurements and the heating and cooling rates were kept constant (10 °C min$^{-1}$). Field-emission SEM (Nova Nano SEM 450 and 230), TEM (Philips CM200 and JEOL JEM-F200), each coupled with an EDS detector system, were used for the characterisation of morphology and element distributions. The bulk samples for SEM transverse-sectional views were cooled in liquid nitrogen before being fractured to prevent re-melting or distorting of the samples, thereby ensuring that their intrinsic microstructures were revealed. The particle samples for SEM and TEM characterisations were first suspended in ethanol and then dropped onto silicon substrates and TEM grids (carbon-coated copper), respectively. During characterising the grain size distribution, we measured both the long axis and short axis of irregular-shape grains, and their averages were used to present the characteristic grain sizes. Raman spectra were collected on a RENISHAW inVia Raman Microscope using a 532 nm laser source. The touch-printed oxide layer samples for the Raman tests were prepared by first heating the $Bi_xSn_{1-x}$ bulk alloys on a hotplate at 400 °C in atmospheric air. When melted, a heated glass slide was used to squeeze liquid metal drop to expose fresh liquid metal surface. A heated silicone substrate was then placed in touch with the freshly oxidised surface and lifted, after which the surface oxide layer was transferred onto the substrate. AFM measurement was performed on a Bruker Dimension Icon AFM system.

**Electrochemical $CO_2$ reduction**. All electrochemical measurements in this study were carried out with a CHI 760E (CH Instruments, Texas) electrochemical workstation using a customised two-compartment gas-tight H-cell, where the compartments were separated by a glass frit to prevent the reduction products formed at the cathode from re-oxidising on the anode. The cathodic compartment of the cell contains a Bi–Sn nano-alloy electrode as the working electrode and a saturated calomel electrode (SCE) as the reference. The anode compartment contains a Pt wire as the counter electrode. The working electrodes were prepared by dispersing 5 mg of Bi–Sn nano-alloy catalysts in a 0.5 mL deionized water and ethanol solution (1:1, v/v) followed by the addition of 25 μL of Nafion solution (Sigma-Aldrich, 99.99%). The mixture was sonicated thoroughly to form a homogeneous ink. The working electrodes were then prepared by drop-casting the catalyst inks onto Teflon-lined carbon fibre paper to achieve a loading of 1 mg cm$^{-2}$. The electrolyte utilised in this study for $CO_2$ reduction was 0.1 M KHCO$_3$. Preceding each experiment, the cathodic compartment of the H-cell was purged with $CO_2$ for 30 min, and the saturated 0.1 M KHCO$_3$ solution gave a pH measurement of 6.8. All potentials measured in this study were converted to the reversible hydrogen electrode (RHE) reference for the purpose of comparison, using the following equation: $E_{RHE}$ (V) $= E_{SCE}$ (V) $+ 0.245 + 0.059 \times$ pH. Potentiostatic studies were carried out at various potentials for a duration of 1 h. Gas chromatograph (Shimadzu, Model 2010 plus), equipped with both a thermal conductivity detector (TCD) and a flame ionisation detector (FID), were used to quantify the gas phase products after 1-h bulk electrolysis. 0.5 mL of liquid sample was collected at the end of each experiment and was mixed with 0.1 mL of $D_2O$ and 7.143 ppm of DMSO as an internal standard and were analysed using a 600 MHz 1H 1D liquid NMR spectrometer (Bruker Advance) at 25 °C. The 1D 1H spectrum was measured under water suppression with a pre-saturation method. The quantities of products were calculated by comparing the integral areas of the observed liquid product with that of the DMSO.

**Photocatalytic dye degradation**. The annealed Bi–Sn nano-alloys for the photocatalysis experiments were prepared by annealing the samples in atmospheric air at 500 °C for 1 h. The eutectic sample was also annealed at different temperatures to investigate the temperature dependence of its photocatalytic activity. The degradation of the annealed samples against Congo red dye was used to evaluate the photocatalytic activity. The concentration of dye and photocatalyst was $1.44 \times 10^{-5}$ mol L$^{-1}$ and 0.09 mg mL$^{-1}$, respectively. During the experiment, 1 simulated solar illumination with an irradiating distance of 0.15 m was used and the solution was stirred constantly at 400 rpm. The degradation rate of the dye was determined by measuring the absorbance of the samples using a UV–Vis–NIR Spectrophotometer (Cary 5000, Agilent Technologies) every 20 min for a 120 min period. The characteristic absorbance peak of Congo red at 500 nm was used for the degradation calculation, with base line corrected.

## Data availability

All relevant data are available from the authors upon reasonable request.

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

## Acknowledgements

The authors would like to acknowledge the Australian Research Council (ARC) Laureate Fellowship grant (FL180100053) and ARC Center of Excellence FLEET (CE170100039) for the financial coverage of this work.

## Author contributions

J.T. and K.K.-Z. initiated the concept and designed the experiments. J.T. conducted the experiments and carried out the characterisations with the help of R.D., M.B.G., S.A.I.-S., A.Z., T.D., J.Y., P.K., S.C. and R.D.T. The following individuals contributed to the data analyses, scientific discussions and authorship of the paper: J.T., R.D., R.B.K., R.A. and K.K.-Z. All authors revised the manuscript and provided helpful comments.

## Competing interests

The authors declare no competing interests.
