## [Peer Review File · Nature Communications]

Reviewers' comments:

Reviewer #1 (Remarks to the Author):

In this manuscript, the authors have synthesized Bi-Sn nano-alloys with different Bi/Sn ratio using the liquid phase ultrasonication from bulk-alloys, and found that the eutectic nano-alloy processes more excellent catalytic performance compared with its counterparts. The authors have described a complete story and found an intriguing phenomenon. However, in consideration of the novelty and depth, the authors may send this manuscript to other more specific journals. Here are the questions and concerns need to be addressed:

1. The authors have stressed advantages of the liquid phase ultrasonication technique on preparing nano-alloys. However, this technique could only apply to low melting-temperature alloys, such as Bi-based alloys. Dealloying and hydrothermal methods can also synthesize nano-alloys without the limitation of temperature. So, what is the distinct advantage of this method?
2. The authors used TEM and spectrum to investigate the structures and the nature of solidification of Bi-Sn nano-alloy systems. In Fig. 5, the contrast in HRTEM cannot represent atomic image, since the contrast from HRTEM is very complex and ambiguous. Thus, the discussion on point defects in Fig. 5l is not accurate. The authors have not provided adequate evidence to prove that there are more defects in the eutectic system compared with others. Besides, the process and nature of solidification have not been discussed in detail in this paper.
3. In Fig. 3i, there is only one melting peak for both of $x=0.4$ and $x=0.57$, are there any other criteria to confirm this alloy is eutectic?
4. In general, the smaller size, the better catalytic activity of nanoparticles. However, the eutectic sample, which is not the smallest, has the best catalytic performance as shown in Fig. 6c. This is the major conclusion of this study. However, the intrinsic mechanism that is responsible for the better catalytic activity is unclear. The referee understand that this study is not deep and novel enough for Nature Communications.

Reviewer #2 (Remarks to the Author):

The authors present a preparation method for bismuth-tin ($\text{Bi}_x\text{Sn}_{1-x}$) bimetallic nanoparticles, discussing the structural properties of several different "bulk alloy" compositions (with a specific eutectic composition of $x=57$ atom%). They demonstrated their materials' performance properties for CO_2 electrocatalytic reduction and the high-temperature, air-calcined versions for photocatalytic oxidation of Congo red in water. They showed that the eutectic composition had the best performance, among all other compositions. The experimental results are good quality, and the analysis is reasonable.

Unfortunately, to this reviewer, this materials system does not effectively serve to illustrate "nanotechnology-enabled metallurgy," as offered by the authors. Only this bimetallic alloy material was presented, suggesting a rather limited materials approach. There were missed opportunities to discuss the thermodynamics the materials; many passing references were made ("the eutectic phase transition thermodynamics", "the governing thermodynamics during solidification", "governed by the Gibb's free energy of oxidation") but there was no deep discussion, giving an impression of superficial referencing. The phase segregation of the $\text{Bi}_x\text{Sn}_{1-x}$ alloys, when nano-sized, was observed, but insufficient attention was given to explain more deeply how the two metals are distributed at the atomic level. A stronger explanation of eutectic melting would have been welcome, for the nanoparticle materials. Ultrasonication is a non-thermodynamic process used to fragment the bulk alloys into nano-sized domains. The authors indicate that shear stresses are the reason, but did not satisfactorily how this is related to temperature and the different melting points of the alloys (eutectic or not).

The presented results did not match the (grandiloquent) writing of the authors, and so this reviewer concludes that a convincing case of "nanotechnology-enabled metallurgy" was not made. The structural data would be interesting to materials scientists interested in catalytic observations.

Specific comments:

* The discussion of heterogeneous catalysis was poor. The authors correctly state that "...evidence shows that bimetallic and multimetallic nano-alloys are superior catalysts relative to their monometallic counterparts^{25,26}." (Line 70), but this does not apply to any arbitrary pair of metals, as the authors imply by this work. They conclude from their TEM work that "The atomic-scale defect containing eutectic Bi-Sn nano alloy, with the highest frequency of point defects, edge and screw dislocations, is therefore expected to be more catalytically active in comparison to its non-eutectic counterparts." (Line 306), which is not a proper assumption either.

* Fig 2 is not that helpful, since the length scale is much larger than those of the nano-alloys.

* It is difficult to ascertain what structures are the nano-versions of the Bi_xSn_{1-x} compositions. Idealized cartoon schematics would have helped summarize their characterization results in a clearer manner.

* The choice of reactions was curious. There was no a priori reason for Bi-Sn to be active for CO₂RR. There was no discussion or evaluation of benchmark CO₂RR materials; the authors should comment on how their best material compares to the commonly studied CO₂RR material. As the authors conclude, the photocatalytic degradation of Congo red was due to Bi₂Sn₂O₇ (and SnO₂) being partially formed after calcining their eutectic nano-alloy powder. Bi₂O₃ was also formed, but it is rarely studied as a photocatalyst, in spite of what the authors imply (line 356). The authors did not explain how the material was better, but wrote that "the results suggest that it may be a general trend for the eutectic ratio to be the optimum for developing catalysts using liquid phase ultrasonication." (line 385)

* Line 334: what is "cocking"?

* The experimental design for this work was incomplete. Control samples were not tested, for example, pure SnO₂, pure Bi₂O₃, conventionally prepared bimetallic materials, benchmark catalysts.

* Figure 7; Line 490: "The degradation rate of the dye was studied" but only degradation % was reported. Properly, the rate constants should have been reported and compared. Percent loss of Congo red is important observe, but does not fully and quantitatively capture the differences in catalyst performance.

Reviewer #3 (Remarks to the Author):

First of all, well written abstract and introduction. If the story can go as described in the abstract, and the assumption made in the introduction can be validated, there is no doubt that this is a Nat Comm level work.

However, after going through the full text, the reviewer is quite disappointed indeed, failing to see what are promised in the abstract and introduction turn into reality. The reviewer does not believe the authors have a convincing story here, and is also challenging the authors' knowledge on eutectic alloys for them to be qualified enough to write a story based on eutectic alloys.

The reviewer lists some main criticism against recommending acceptance of this work.

1. They talk too much, almost unnecessarily, about the basics of eutectic alloys, and use the wording like "Strikingly, when the Bi-Sn ratio reaches the eutectic value ($x = 0.57$), the alternately arranged Bi and Sn lamellae become dominant, ruling out the discrete growth regime as observed in the

hypoeutectic samples." Well, this is nothing striking at all to people knowing what eutectic alloys mean. This also relates to the description on the DSC behaviors. Nevertheless, these statements are basically unnecessary, but not wrong. They then start to go wrong afterwards, starting from using "split of the DSC peaks" to describe what is seen in Fig.3, then to "shoulders of the non-eutectic samples are shaved after the liquid phase ultrasonication, indicating accompanying compositional changes", and finally to "the single melting peak of the eutectic sample implies that the eutectic Bi-Sn ratio is still maintained after ultrasonication". At this point, the reviewer already lost the interest to continue the reading. Apparently, they do not know much about eutectic alloys, and it seems that they also do not know much about thermal measurements.

2. They made such a strange statement in Page 9: This indicates that under the current liquid phase ultrasonication conditions, the particle size is not determined by the melting point of the bulk samples, instead, it correlates with the Bi-Sn mixing ratio. So for them, the melting point does not change with the Bi-Sn mixing ratio? The reviewer even wants to challenge their knowledge on thermodynamics now.

3. In Page 12, quite annoyingly, they started to describe the co-existence of Bi oxide, immediately after they just explained why Bi oxide is absent.

4. Most importantly, the reviewer does not see eutectic nano-alloys are much superior to non-eutectic nano-alloys, regarding the catalytic and photocatalytic activity, from what they show in Fig. 6 and Fig. 7, and also does not see the claimed evidence for this so-called superiority, which is the enhanced defects. This is the kernel of this story, and it is rather weakly supported.

5. Page 20: The generation of $\text{Bi}_2\text{Sn}_2\text{O}_7$, also a visible-light-driven photocatalyst, indicates the fine mixing of Bi and Sn. The reviewer sees no such a connection that can be made here.

Responses to Reviewer #1:

Comment: In this manuscript, the authors have synthesized Bi-Sn nano-alloys with different Bi/Sn ratio using the liquid phase ultrasonication from bulk-alloys and found that the eutectic nano-alloy processes more excellent catalytic performance compared with its counterparts. The authors have described a complete story and found an intriguing phenomenon. However, in consideration of the novelty and depth, the authors may send this manuscript to other more specific journals. Here are the questions and concerns need to be addressed:

Response: Firstly, we would like to sincerely thank you for reviewing our work, your support and constructive advice for improving the work. After carefully reading your comments, we revised our manuscript accordingly. We added more measurements and in-depth discussions to further strengthen the conclusions. In what follows we present our point-by-point responses to your comments.

Comment: 1. The authors have stressed advantages of the liquid phase ultrasonication technique on preparing nano-alloys. However, this technique could only apply to low melting-temperature alloys, such as Bi-based alloys. Dealloying and hydrothermal methods can also synthesize nano-alloys without the limitation of temperature. So, what is the distinct advantage of this method?

Response: Thanks for the comments. To address your important comments, we present the following discussions and we also added a few extra sections in both main manuscript and Supporting Information. The main advantages of the liquid phase ultrasonication method include:

1) It is a physical approach which directly uses bulk metals as precursors to produce nanomaterials. Different from the dealloying and hydrothermal methods, “Ultrasonication process is pursued for creating catalytic nano-alloys as it offers high yield and low cost, and also avoid unwanted by-products that are formed in many typical chemical/electrochemical reactions.” The quotation was added to the paper.

2) In the ultrasonication process, the samples are in their liquid states during preparation and are solidified afterwards. This allows two fundamental processes to take place, namely tuning of the surface composition for selective activity and crystal phase engineering in the bulk of the nanoparticles during solidification. During ultrasonication, many of the parameters can be tightly controlled to achieve desired structure and functionality. Examples of readily controllable parameters, which influence of surface

oxidation and solidification, are the environment and heating/cooling rate, which will be the focus of our follow-up works. Some discussions regarding these advantages were added in various sections of the paper.

3) Regarding the types of alloys feasible for the liquid phase ultrasonication method, we note that there indeed exists plenty of alloy systems to choose from. This is guaranteed by the following facts:

a) Post transition metals and zinc group metals and their alloys can be processed using this technique at below about 450°C. Exclusion of Zn bring this to about 350°C. An explanation about this was added to the revised manuscript.

b) The incorporation of different metals forms alloys with lower melting point than the starting metals is another feasible strategy. Many high-melting-point metals can be incorporated into low-melting-point ones and the resulted melting temperature of the alloys is still be lower than that of all the precursor metals. In the added Table R1, we list examples of some binary eutectic alloy systems which could be processed using ultrasonication. Note that in the table we restrict the melting temperature below 300°C and the alloys to binary systems. Many more binary alloy systems are available either at higher melting temperature and ternary, quaternary alloy systems and beyond should also be considered. Discussions about this matter were added in different sections of the paper.

c) It has been demonstrated that the ultrasonication can be performed at temperature as high as 700°C, so the ultrasonication technology is not likely to be a major limitation for the method [*Nature* 528, 539 (2015)]. Also, there are readily available solvent options (such as DMSO and silicone oil used in this study and others like glycerol) to reach operation temperatures as high as 400°C. Ionic liquids and inorganic molten salts can be used for conditions which require higher operation temperature.

d) Importantly when incorporated into alloys, different metals compete to attain at the surface of liquid alloys. This means even fractional amount of metal in alloys can still play a big role in chemical/electrochemical reactions given the selective migration of metal species to the surface (Questioning its underlying mechanisms here also rises fundamental interests). For instance, it is found that adding a few weight percent of cerium (Ce, melting point 799°C) to Galinstan (an alloy containing 68.5 wt% gallium, 21.5 wt% indium and 10 wt% tin, melting point:

13.2°C) causes Ce to enrich at the surface, which changes the activity of the alloy towards CO₂ reduction and enables producing solid carbon materials at room temperature [*Nat. Commun.* 10, 865 (2019)].

The following paragraphs were added in the paper:

“Although low temperature melting point metals, including zinc group and post transition metals, are seen to be the first beneficiary of the demonstrated procedure, the concept can also be extended to many other metals. This includes selected high-melting-point metals that can be incorporated in low-melting-point ones to produce alloys with significantly lowered melting temperature (see Supplementary Table 2 for a list of binary systems). Additionally, when incorporated, different metals will compete to attain the dominance on the surface of the alloys. This means that if chosen correctly fractional amount of a metal in alloys can still play a significant role in chemical or catalytic reactions that rely on surface properties.”

The following Table was also added to the Supplementary Information:

Table A1. Eutectic composition and melting point of some binary alloy systems.

Alloy system	Eutectic composition	Melting point of A	Melting point of B	Eutectic melting point of A-B (°C)
A-B	A_{wt%}B_{wt%}	(°C)	(°C)	
Bi-Cd	Bi ₆₀ Cd ₄₀	271.4	321.1	146
Bi-In	Bi _{43.3} In _{56.7}	271.4	156.6	72.7
Bi-Li	Bi ₂₃ Li ₇₇	271.4	180.6	175.0
Bi-Pb	Bi _{55.2} Pb _{44.8}	271.4	327.5	125.5
Bi-Pd	Bi ₉₇ Pd ₃	271.4	1555	256
Bi-Pt	Bi _{99.2} Pt _{0.8}	271.4	1769	259
Bi-Sm	Bi ₉₉ Sm ₁	271.4	1074	252
Bi-Sn	Bi ₅₇ Sn ₄₃	271.4	232.0	139
Bi-Te	Bi _{98.3} Te _{1.7}	271.4	449.6	266
Bi-Yb	Bi ₉₅ Yb ₅	271.4	819	250
Bi-Zn	Bi _{97.3} Zn _{2.7}	271.4	419.6	254.5
Cd-In	Cd _{25.3} In _{74.4}	321.1	156.6	126
Cd-Pb	Cd _{17.5} Pb _{82.5}	321.1	327.5	246
Cd-Sb	Cd ₉₂ Sb ₈	321.1	630.8	290
Cd-Sn	Cd _{32.25} Sn _{67.75}	321.1	232.0	176
Cd-Tl	Cd ₁₇ Tl ₈₃	321.1	304	203.5
Cd-Zn	Cd _{82.6} Zn _{17.4}	321.1	419.6	266
Cu-In	Cu _{0.9} In _{99.1}	1064.9	156.6	153
Cu-Tin	Cu _{0.7} Sn _{99.3}	1064.9	232.0	227

Dy-Sn	Dy _{1.2} Sn _{98.8}	1412	232.0	215
Ga-Ag	Ga _{94.5} Ag _{5.5}	29.8	951.9	25.0
Ga-In	Ga _{78.6} In _{21.4}	29.8	156.6	15.3
Ga-Mn	Ga ₉₉ Mn ₁	29.8	1246	29.8
Ga-Sn	Ga _{86.5} Sn _{13.5}	29.8	232.0	20.5
Ga-Yb	Ga ₉₉ Yb ₁	29.8	819	27
Ga-Zn	Ga _{96.4} Zn _{3.4}	29.8	419.6	24.7
Li-Pd	Li ₅₁ Pd ₄₉	180.6	1555	145
Li-Sn	Li ₉₈ Sn ₂	180.6	232.0	179
Li-Sr	Li ₃₇ Sr ₆₃	180.6	769	134
Li-Tl	Li ₇₇ Tl ₂₃	180.6	304	175
Li-Zn	Li _{69.3} Zn _{30.7}	180.6	419.6	161
Mg-Sn	Mg _{2.1} Sn _{97.9}	650	232.0	203.5
Mg-Tl	Mg ₃ Tl ₉₇	650	304	202
Pb-Pd	Pb _{95.5} Pd _{4.5}	327.5	1555	260
Pb-Pt	Pb ₉₅ Pt ₅	327.5	1769	290
Pb-Sb	Pb _{88.9} Sb _{11.1}	327.5	530.8	251.7
Pb-Sn	Pb _{38.1} Sn _{61.9}	327.5	232.0	183
Pd-Sn	Pd ₉₉ Sn ₁	1555	232	230
Pd-Tl	Pd ₉₉ Tl ₁	1555	302	293
Pt-Sn	Pt _{0.8} Sn _{99.2}	1769	232.0	226
Sb-Tl	Sb _{19.7} Tl _{80.3}	630.8	304	195
Se-Tl	Se _{47.5} Tl _{52.5}	221	304	199
Sn-Sr	Sn ₉₉ Sr ₁	232.0	759	230
Sn-Tl	Sn ₅₇ Tl ₄₃	232.0	304	166
Sn-Zn	Sn _{91.2} Zn _{8.8}	232.0	419.6	198.5
Te-Tl	Te ₄₂ Tl ₅₈	449.6	304	224
Tl-Zn	Tl ₉₇ Zn ₃	304	419.6	292

Comment: 2. The authors used TEM and spectrum to investigate the structures and the nature of solidification of Bi-Sn nano-alloy systems. In Fig. 5, the contrast in HRTEM cannot represent atomic image, since the contrast from HRTEM is very complex and ambiguous. Thus, the discussion on point defects in Fig. 5l is not accurate. The authors have not provided adequate evidence to prove that there are more defects in the eutectic system compared with others. Besides, the process and nature of solidification have not been discussed in detail in this paper.

Response: Thanks for raising this comment. After carefully considering your advice, we took many extra TEM images, added extra analysis and made significant changes to the TEM characterization part to better support our discussions. We obtained more HRTEM images for different Bi-Sn nano-alloy samples. According to different orientations of the lattice fringes, we outlined the grain boundaries, based on which we were able to evaluate the grain size of the nano-alloys. In doing so, we measured both the long axis and short axis of irregular-shape grains and their average value was used for presenting the characteristic grain sizes. The additional grain size distributions of different samples were included in the inset of Fig. A1e-h and they indeed show that the eutectic sample has the smallest grain size in comparison with the non-eutectic ones. Grain boundaries are defects which are beneficial for enhancing catalysis [*J. Am. Chem. Soc.* 137, 4606-4609 (2015)]. For example, the binding energy of the reaction intermediates in CO₂ reduction reaction can be tuned near these defect sites due to local spatial symmetry breaking [*Angew. Chem. Int. Ed.* 56, 3645-3649 (2017)]. Therefore, the eutectic nano-alloy samples with the smallest grain sizes (and not the smallest particle size) can provide more defects sites and demonstrates the highest selectivity and current density among all the samples during the CO₂ reduction reaction.

The main manuscript was edited very carefully to include the above-discussion in detail. The followings are the added figures to the paper:

Fig. A1 Crystallographic characterization of the $\text{Bi}_x\text{Sn}_{1-x}$ nano-alloys. **a-d** Examples of DF-TEM images of individual particles of different $\text{Bi}_x\text{Sn}_{1-x}$ nano-alloy samples. The insets present the FFT patterns of a $40 \text{ nm} \times 40 \text{ nm}$ region of each particle as indicated by the dash-line boxes. **e-h** BF-HR-TEM images of the nano-alloys with their grain boundaries outlined. The insets present their respective grain size distribution. **i-k** BF-HR-TEM images showing extrinsic point defects indicated by arrows (**i**), and line defects (edge dislocations) indicated by T-shape symbols (**j**, **k**). **l** TEM of a eutectic nano-alloy particle and an extracted illustration shows its heterostructures based on the characterization results.

Another evidence that shows the eutectic sample has more defects than the non-eutectics can be found by comparing the SnO₂ defect Raman mode of the annealed samples. SnO₂ is known to have a defect Raman mode at 577 cm⁻¹ and a nearby active Raman mode at 633 cm⁻¹ [*J. Phys. Chem. C* 115, 118-124 (2010)]. As shown in Fig. A2a, the annealed eutectic sample has the highest relative intensity of SnO₂ defects (characterized as I_D/I_A), where I_D and I_A are the intensity of the SnO₂ defect and active Raman mode, respectively. Moreover, as shown in Fig. A2b, there is a good correlation between the relative defect intensity (I_D/I_A) and rate constant k of the photocatalytic dye degradation, which further confirms that the highest amount of defect of the eutectic sample is primarily responsible for its enhanced catalytic activity.

The formation of more defects in the eutectic nano-alloy can be explained as follows that is now added to the paper: “The formation of defects implies stress build-up in individual nanoparticles [*Acta Mat.* 55, 1241-1254 (2007)]. When cooled down, the non-eutectic nano-alloys experience gradual and successive growth of different phases and a special liquid-solid state transient coexistence depending on their compositions. By contrast, for eutectic nano-alloy particles, different phases crystallize simultaneously at the same point during the same cooling condition. Such rapid yet simultaneous crystallization does not allow stress to dissipate within individual particles. Therefore, more defects can be formed during the solidification of the eutectic nano-alloys.”

Regarding your concerns about point defect discussion, we agree that the HRTEM images cannot represent atomic images. We removed the previous figures and provided another image, Fig. A1i, which better shows the locally distorted lattice structures that is supposed to be induced by substitution of atomic species. Our responses to your comment here have also been added to the revised manuscript accordingly to strengthen the discussions.

Fig. A2 a Relative Raman intensity of the SnO₂ defect mode I_D (577 cm⁻¹) and active mode I_A (633 cm⁻¹). **b** The dependence of SnO₂ defect intensity (characterised by I_D/I_A) and dye degradation rate constant k on the mixing ratio of the Bi_xSn_{1-x} nano-alloy samples.

Comment: 3. In Fig. 3i, there is only one melting peak for both of $x=0.4$ and $x=0.57$, are there any other criteria to confirm this alloy is eutectic?

Response: Thanks, and we understand your comment. We included extra figures and explanations to address it.

Eutectic system is defined by its single-temperature phase transition behaviours and DSC measurement is the most conventional method for determining the occurrence of eutectic state and phase transition behaviours. As indicated in the inset of Fig. 3k of the revised manuscript (plotted in Fig. A3 below, black curves), only the eutectic nano-alloy sample (the small particles) shows a single melting peak. The shoulder of the melting peak for the two samples $x = 0.20$ and $x = 0.80$ is obvious and the relatively smooth transition of the sample $x = 0.40$ may cause your concern.

Fig. A3 covers a wide temperature range from about 145°C to 225°C. As can be seen, the melting of $x = 0.40$ sample takes places at a temperature away from the main peak and therefore it should be non-eutectic. This can also be better illustrated using the temperature derivative of heat flow as also shown in Fig. A3 (red curves). The temperature derivatives of heat flows of all non-eutectic samples show temperature-dependent variations (the $x = 0.40$

sample shows a significant slope) while that of the eutectic sample $x = 0.57$ remains constant and flat. Therefore, it can be confirmed that only the $x = 0.57$ nano-alloy sample is eutectic. To address the comment, the extra Fig. A3 was added to the Supplementary Information.

Fig. A3 Plots of heat flows (black curves) and their temperature derivatives (red curves) for different Bi-Sn nano-alloy samples shown in the insets of Fig. 3i of the manuscript.

Comment: 4. In general, the smaller size, the better catalytic activity of nanoparticles. However, the eutectic sample, which is not the smallest, has the best catalytic performance as shown in Fig. 6c. This is the major conclusion of this study. However, the intrinsic mechanism that is responsible for the better catalytic activity is unclear.

Response: Thanks for the comment. We included significant number of measurements and discussions to address your comment. The explanations regarding the extra materials added are as follows:

(1) As discussed in our work, the particle size of the alloys does not play a significant role in determining the CO₂RR activity. In our study, increasing Bi content ($x = 0.20$ to 0.80)

within the nano-alloy leads to a decrease in particle size whereas the partial current density for formate (j_{HCOO^-}) increases till $x=0.57$ before declining again, indicating that there is no direct correlation between particle size and CO_2RR activity. The same trend is also seen in the case of photocatalysis.

We added more measurements and discussions to show that the electrocatalytic activity of the nanoalloys are correlated to the intensity of defects present in the catalysts. It is known that the introduction of defects in electrocatalysts improve CO_2RR [*Angew. Chem. Int. Ed.* 57, 6054 (2018); *ACS Sustain. Chem. Eng.* 6, 1670 (2018)] as the defect sites govern the adsorption of CO_2 reactants and can also stabilize the formate anion radical intermediates [*Nat. Commun.* 8, 14503 (2017)].

To Address your comment, we carefully added extra figures (Fig. A1) and analysed the frequency and dimensions of the grains within nanoparticles. It can be observed from Fig. A1e-h that the eutectic sample has the smallest grain size in comparison to the non-eutectic ones. This correlates well with our experimental findings as the eutectic nano-alloy with the highest amount of grain boundaries, vacancies and dislocations demonstrate the highest selectivity and j_{HCOO^-} . This conclusion is further supported by Raman spectroscopy measurements as presented in Fig. A2.

(2) Another important point that we endeavoured to further highlight in the new version of the paper is the solidification process of the bulk and nano-alloys from liquid metals. We show similar trend in the solidification of metals in the core of nanoparticles and the bulk of liquid metal. However, we investigated the matter further as what is formed on the surface of nano-alloys in terms of the types of oxides and how they are different from the surface of the bulk.

(3) We further highlighted that the method can go beyond the example of Sn-Bi example. In the new version of the paper we explained that a similar process can be applied on zinc group and post transition metals. Additionally, we added a table of possible binary compounds, with components in transition metals, suitable for the process

To summarize, our study proposes a low cost and scalable method for generating nano-alloy catalysts by taking advantage of the melting temperature drop during the alloying process. It demonstrates the feasibility of the strategy and reveals fundamental nanoscale phase separation and selective oxidation processes involved during ultrasonication and solidification. Following your kind suggestions, the advantages of the method compared to other processes and the feasibility of extending it to other alloy systems are also rationalized. The catalytic activity of the obtained nano-alloys is evaluated using two different catalytic

processes, both showing superior activity for the eutectic sample to provide evidence that the conclusion presented in the paper is rigorous. This is a favourable finding for liquid metal-based nanotechnology. By further characterizing the grain size distribution of the nano-alloy samples, we show the eutectic sample has more defects than its non-eutectic counterparts. Additionally, we further characterize the defects using Raman spectroscopy and show that the annealed eutectic sample also has the highest SnO₂ defect mode intensity. We further present good correlation between the defect intensity and catalytic activity of the samples. Therefore, we attribute the superior performance of the eutectic sample to its more predominant defect formation than the non-eutectics, which is a result of the rapid yet simultaneous eutectic phase transition behaviour.

Responses to Reviewer #2:

Comment: The authors present a preparation method for bismuth-tin ($\text{Bi}_x\text{Sn}_{1-x}$) bimetallic nanoparticles, discussing the structural properties of several different "bulk alloy" compositions (with a specific eutectic composition of $x=57$ atom%). They demonstrated their materials' performance properties for CO_2 electrocatalytic reduction and the high-temperature, air-calcined versions for photocatalytic oxidation of Congo red in water. They showed that the eutectic composition had the best performance, among all other compositions. The experimental results are good quality, and the analysis is reasonable.

Unfortunately, to this reviewer, this materials system does not effectively serve to illustrate "nanotechnology-enabled metallurgy," as offered by the authors. Only this bimetallic alloy material was presented, suggesting a rather limited materials approach. There were missed opportunities to discuss the thermodynamics the materials; many passing references were made ("the eutectic phase transition thermodynamics", "the governing thermodynamics during solidification", "governed by the Gibb's free energy of oxidation") but there was no deep discussion, giving an impression of superficial referencing. The phase segregation of the $\text{Bi}_x\text{Sn}_{1-x}$ alloys, when nano-sized, was observed, but insufficient attention was given to explain more deeply how the two metals are distributed at the atomic level. A stronger explanation of eutectic melting would have been welcome, for the nanoparticle materials. Ultrasonication is a non-thermodynamic process used to fragment the bulk alloys into nano-sized domains. The authors indicate that shear stresses are the reason but did not satisfactorily how this is related to temperature and the different melting points of the alloys (eutectic or not).

The presented results did not match the (grandiloquent) writing of the authors, and so this reviewer concludes that a convincing case of "nanotechnology-enabled metallurgy" was not made. The structural data would be interesting to materials scientists interested in catalytic observations.

Response: Thank you very much for your helpful review comments. We totally understood and value your comments and made a significant effort to address them. In response, the paper was significantly modified, and several new measurements were included. Bearing your comments in mind, in what follows, we present more results and in-depth discussions about: (1) accessible alloy systems of our strategy, (2) thermodynamics of the materials, (3) phase distribution, and (4) eutectic phase transformation to better support our conclusions and improve the work. (5) Additionally, we give explanations to the

compositional difference of the nano-alloys produced by ultrasonication. Changes were made to the revised manuscript accordingly and all these alterations were highlighted for your review convenience. We also modified the introduction to more directly highlight the content and novelty of the work.

(1) Accessible alloy systems of our strategy: In this study, as a proof-of-concept, we select the bismuth-tin (Bi-Sn) alloy as the input for the system since this binary alloy shows no intermetallic phase, which makes the study less complex and more informative. Regarding the types of alloys feasible for the liquid phase ultrasonication method, we note that there indeed exists plenty of alloy systems to choose from. In order to assure that we have addressed your important comment, we present the following discussions and also added a few extra sections in both main manuscript and Supporting Information.

The main advantages of the liquid phase ultrasonication method include:

1) It is a physical approach which directly uses bulk metals as precursors to produce nanomaterials. Different from the dealloying and hydrothermal methods, “Ultrasonication process is pursued for creating catalytic nano-alloys as it offers high yield and low cost, and also avoid unwanted by-products that are formed in many typical chemical/electrochemical reactions.” The quotation was added to the paper.

2) In the ultrasonication process, the samples are in their liquid states during preparation and are solidified afterwards. This allows two fundamental processes to take place, namely tuning the surface for selective activity and crystal phase engineering in the bulk of the nanoparticles during solidification. During ultrasonication, many of the parameters can be tightly controlled to achieve desired structure and functionality. Examples of readily controllable parameters, which influence of surface oxidation and solidification, are the environment and heating/cooling rate, which will be the focus of our follow-up works. Some discussions regarding these advantages were added in various sections of the paper.

3) Regarding the types of alloys feasible for the liquid phase ultrasonication method, we note that there indeed exists plenty of alloy systems to choose from. This is guaranteed by the following facts:

a) Post transition metals and zinc group metals and their alloys can be processed using this technique at below about 450°C. Exclusion of Zn bring this to below about 350°C.

b) The incorporation of different metals forms alloys with lower melting point than the starting metals is another feasible strategy. Many high-melting-point metals

can be incorporated into low-melting-point ones and the resulted melting temperature of the alloys is still be lower than that of all the precursor metals. In the added Table, we list examples of some binary eutectic alloy systems which could be processed using ultrasonication. Note that in the table we restrict the melting temperature below 300°C and the alloys to binary systems. Many more alloy binary systems are available either at higher melting temperature and also ternary, quaternary alloy systems and beyond should also be considered.

c) It has been demonstrated that the ultrasonication can be performed at temperature as high as 700°C, so the ultrasonication technology is not likely to be a major limitation for the method [*Nature* 528, 539 (2015)]. Also, there are readily available solvent options (such as DMSO and silicone oil used in this study and others like glycerol) to reach operation temperatures as high as 400°C. Ionic liquids and inorganic molten salts can be used for conditions which require higher operation temperature.

d) Importantly, when incorporated, different metals will compete to attain at the surface of liquid alloys. This means even fractional amount of metal in alloys can still play a big role in chemical/electrochemical reactions given the selective migration of metal species to the surface (Questioning its underlying mechanisms here also rises fundamental interests). For instance, it is found that adding a few weight percent of cerium (Ce, melting point 799°C) to Galinstan (an alloy containing 68.5 wt% gallium, 21.5 wt% indium and 10 wt% tin, melting point: 13.2°C) causes Ce to enrich at the surface, which changes the activity of the alloy towards CO₂ reduction and enables producing solid carbon materials at room temperature [*Nature Communications* 10, 865 (2019)].

To address your comments, many paragraphs were added and edited in the body of the manuscript. Especially, the following was added to the paper:

“Although low temperature melting point metals, including zinc group and post transition metals, are seen to be the first beneficiary of the demonstrated procedure, the concept can also be extended to many other metals. This includes selected high-melting-point metals that can be incorporated in low-melting-point ones to produce alloys with significantly lowered melting temperature (see Supplementary Table 2 for a list of binary systems). Additionally, when incorporated, different metals will compete to attain the dominance on the surface of the alloys. This

means that if chosen correctly fractional amount of a metal in alloys can still play a significant role in chemical or catalytic reactions that rely on surface properties.”

The following Table was also added to the Supplementary Information:

Table B1. Eutectic composition and melting point of a few binary alloy systems.

Alloy system A-B	Eutectic composition A _{wt%} B _{wt%}	Melting point of A (°C)	Melting point of B (°C)	Eutectic melting point of A-B (°C)
Bi-Cd	Bi ₆₀ Cd ₄₀	271.4	321.1	146
Bi-In	Bi _{43.3} In _{56.7}	271.4	156.6	72.7
Bi-Li	Bi ₂₃ Li ₇₇	271.4	180.6	175.0
Bi-Pb	Bi _{55.2} Pb _{44.8}	271.4	327.5	125.5
Bi-Pd	Bi ₉₇ Pd ₃	271.4	1555	256
Bi-Pt	Bi _{99.2} Pt _{0.8}	271.4	1769	259
Bi-Sm	Bi ₉₉ Sm ₁	271.4	1074	252
Bi-Sn	Bi ₅₇ Sn ₄₃	271.4	232.0	139
Bi-Te	Bi _{98.3} Te _{1.7}	271.4	449.6	266
Bi-Yb	Bi ₉₅ Yb ₅	271.4	819	250
Bi-Zn	Bi _{97.3} Zn _{2.7}	271.4	419.6	254.5
Cd-In	Cd _{25.3} In _{74.4}	321.1	156.6	126
Cd-Pb	Cd _{17.5} Pb _{82.5}	321.1	327.5	246
Cd-Sb	Cd ₉₂ Sb ₈	321.1	630.8	290
Cd-Sn	Cd _{32.25} Sn _{67.75}	321.1	232.0	176
Cd-Tl	Cd ₁₇ Tl ₈₃	321.1	304	203.5
Cd-Zn	Cd _{82.6} Zn _{17.4}	321.1	419.6	266
Cu-In	Cu _{0.9} In _{99.1}	1064.9	156.6	153
Cu-Tin	Cu _{0.7} Sn _{99.3}	1064.9	232.0	227
Dy-Sn	Dy _{1.2} Sn _{98.8}	1412	232.0	215
Ga-Ag	Ga _{94.5} Ag _{5.5}	29.8	951.9	25.0
Ga-In	Ga _{78.6} In _{21.4}	29.8	156.6	15.3
Ga-Mn	Ga ₉₉ Mn ₁	29.8	1246	29.8
Ga-Sn	Ga _{86.5} Sn _{13.5}	29.8	232.0	20.5
Ga-Yb	Ga ₉₉ Yb ₁	29.8	819	27
Ga-Zn	Ga _{96.4} Zn _{3.4}	29.8	419.6	24.7
Li-Pd	Li ₅₁ Pd ₄₉	180.6	1555	145
Li-Sn	Li ₉₈ Sn ₂	180.6	232.0	179
Li-Sr	Li ₃₇ Sr ₆₃	180.6	769	134
Li-Tl	Li ₇₇ Tl ₂₃	180.6	304	175
Li-Zn	Li _{69.3} Zn _{30.7}	180.6	419.6	161

Mg-Sn	Mg _{2.1} Sn _{97.9}	650	232.0	203.5
Mg-Tl	Mg ₃ Tl ₉₇	650	304	202
Pb-Pd	Pb _{95.5} Pd _{4.5}	327.5	1555	260
Pb-Pt	Pb ₉₅ Pt ₅	327.5	1769	290
Pb-Sb	Pb _{88.9} Sb _{11.1}	327.5	530.8	251.7
Pb-Sn	Pb _{38.1} Sn _{61.9}	327.5	232.0	183
Pd-Sn	Pd ₉₉ Sn ₁	1555	232	230
Pd-Tl	Pd ₉₉ Tl ₁	1555	302	293
Pt-Sn	Pt _{0.8} Sn _{99.2}	1769	232.0	226
Sb-Tl	Sb _{19.7} Tl _{80.3}	630.8	304	195
Se-Tl	Se _{47.5} Tl _{52.5}	221	304	199
Sn-Sr	Sn ₉₉ Sr ₁	232.0	759	230
Sn-Tl	Sn ₅₇ Tl ₄₃	232.0	304	166
Sn-Zn	Sn _{91.2} Zn _{8.8}	232.0	419.6	198.5
Te-Tl	Te ₄₂ Tl ₅₈	449.6	304	224
Tl-Zn	Tl ₉₇ Zn ₃	304	419.6	292

(2) Material thermodynamics: The discussion on the thermodynamics of the materials was further strengthened in our revised manuscript with the support of thermodynamic calculations for the Bi-Sn-O system. The FACT-Sage software was used for predicting the formation of possible phases at different compositions in a given system under certain conditions, based on the Gibbs free energy of different phases [FACT-Sage 7.2: Based on FactSage Thermochemical Software and Databases, 2010-2016, *Calphad*, 2016]. The Bi-Sn-O phase diagram under 300 °C and 1 atm, which is close to the ultrasonication conditions, is shown in Fig. B1 below. It can be seen that at SnO is formed preferentially in low-oxygen conditions (region i) and the tendency of formation in the Bi-Sn-O system should be SnO > SnO₂ > Bi₂O₃. The results based on Gibbs free energy agree well with our experimental findings as well as the reported SnO formation conditions [*Thermochimica Acta* 403, 275–285 (2003)]. However, in the ultrasonication case the competition between the metal oxides changes due to the added energy to the system.

Fig. B1 has added to the revised manuscript in Fig. 2.

A comprehensive discussion was also added to the body of the manuscript as follows:

“In the Bi-Sn binary alloy system, the oxidation of the liquid metal surface competes between the formation of SnO, SnO₂, and Bi₂O₃. This is assumed to be governed by the Gibb’s free energy of oxidation when no sonication is applied¹⁵. To assess this assumption, we exfoliate the surface oxide

layers formed on different liquid $\text{Bi}_x\text{Sn}_{1-x}$ bulk samples to examine which oxide phase is selectively formed with no externally applied energy (Fig. 2h and Supplementary Fig. 2). The Raman spectra of the surface oxide layers (Fig. 2i) show that the result of the competition of oxide formation on the surface of bulk Bi-Sn alloy is preferentially won by SnO. Here Bi_2O_3 layer is rarely seen, even at high concentrations of Bi, and SnO_2 only shows itself at $x = 0.80$. The favourability of SnO formation on the surface of bulk is also validated through thermodynamic calculation using FACT-Sage³⁶. As can be seen from Fig. 2j, the formation of SnO can be predicted from the Bi-Sn-O phase diagram at low O_2 concentration (region i) which is matched by our experimental conditions. In addition, the formation of SnO_2 at low Sn ratio (regions ii and iii) in the system can be inferred from the phase diagram.”

“One important observation from these experiments is the difference between the surface oxidation of the bulk and oxides on nano-alloy surface. This deviation is presumably caused by the sonication process which promotes phase separation at the same time as the surface oxidation takes place. We have already discussed about the surface domination of the bulk sample by SnO when no sonication is applied. However, our characterizations revealed that under ultrasonication SnO_2 also appears near the surface when Sn concentration reaches the eutectic value or smaller. For these concentrations the emergence of SnO_2 is favoured, especially when oxygen can be continuously dissolved in the environment, which is the case for the sonication process. Sonication also applies energy to the system and as such can cause phase separation, which allows the emergence of Bi_2O_3 .”

Fig. B1 Phase diagram of the Bi-Sn-O system calculated at 300 °C, 1 atm.

(3) Phase distribution at atomic scale: To further address this issue, extra STEM image and EDX elemental mapping were taken and examples were added to the paper. It can be seen from Fig. B2 that the Bi and Sn phases are distributed uniformly in the nano-alloys. This

seems to be different from that of bulk phase separation in which Bi and Sn form large separated domains after the sonification (Fig. 2 of the manuscript). However, it can be inferred from the XRD results of the nano-alloys that phase separation still exists at nanoscale, which is due to the fact that the Bi-Sn binary system does not form intermetallic phases. These combined results mean that the scale of phase separation in the nano-alloys is beyond resolution of STEM-EDX mapping.

We then additionally surveyed the grain size distribution of different polycrystalline nano-alloy samples (Fig. B3e-h) and found that the grain size (median) for the non-eutectic nano-alloys is about 10 nm and for the eutectic sample is 7.4 nm. Note that the grains can acquire different phases or different crystal orientations of the same phase. Nevertheless, the observations indicate that different phases in the particles can separate into few-nanometre domains. Moreover, even though Bi and Sn form no intermetallic phase, their separation in atomic scales result in extrinsic point defects at these scales. One of such scenarios is shown in Fig. B3i.

Fig. B2 **a** STEM image and EDX element mapping showing the distribution of Bi, Sn and O in a single eutectic Bi-Sn nano-alloy particles. **b** TEM image and EDX elemental mapping showing the distribution of Bi, Sn and O in multiple eutectic Bi-Sn nano-alloy particles.

The following was significantly edited, and many parts were also added to it:

“Fast Fourier transformation (FFT) leads to more pronounced omnidirectionally dispersed patterns for the eutectic sample, indicating more enhanced spatial misorientation of grains within the eutectic nano-alloys in comparison with the non-eutectic nanoparticles. These results mean that, after solidification, phase separation imposes itself more delicately on the eutectic nano-alloys. As shown in Fig. 5e-h, we can further outline the grain boundaries of the bright-field high-resolution TEM (BF-HR-TEM) images taken from different nano-alloy samples according to various orientations of crystal lattices of different grains. Then the size of the grains within individual nanoparticles can be statistically evaluated. The surveys of the grain size distribution of different samples reveal that the overall grain size of the eutectic sample is the smallest among all the samples (insets of Fig. 5e-h), while its particle size is not (Fig. 3e-h). Crystal grains contain two-dimensional defects. Logically the eutectic sample with the smallest grains should have the largest amount of their boundaries and interfaces. We note that the higher polycrystallinity (smaller grain size) in the eutectic nano-alloys is typically found in the HR-TEM images (Supplementary Fig. 8).

In addition, the formation of low-dimensional defects is frequently observed in the eutectic nano-alloys (Fig. 5i-k), which is either absent or rarely seen in the non-eutectic particles. For instance, the locally distorted lattice structures shown in Fig. 5i are likely induced by the substitution of atom species, especially for the oxides, which can be classified as zero-dimensional point defects. Classic one-dimensional edge dislocations are also identified (Fig. 5j, k) at much higher frequencies of occurrence from the eutectic nano-nanoalloys. It can also be seen from Fig. 5j that due to the small scale of grains and the coexistence of local edge dislocations, their glide planes are terminated shortly. Fig. 5k further shows more complex scenarios in which long-range lattice misalignments occur on both sides of the indicated edge dislocation. Presumably, this is formed as screw dislocations set in and couple with the edge dislocations (mixed dislocations)⁴⁶, or by faulted stacking of different atom species. Based on all the characterizations of the nano-alloys, an illustration of their characteristic structure is presented in Fig. 5l. The nano-alloys have a surface containing SnO and Bi₂O₃, a deeper region mostly composed of SnO and SnO₂, and a Bi-Sn metallic core. All these regions feature intense defects.

Defect formation is known to be facilitated in multi-metal systems due to differences in radius and electron structure of the atoms⁴⁷. The Bi-Sn nano-alloys prepared by liquid phase sonication are expected to have enhanced defect formation since, besides the incorporation of different atom species, their crystal structures are established through a phase transition process³⁵. For the purpose of comparison, we note that nano-alloys grown by other methods usually feature ‘near-perfect’ crystal structures or large grain size-to-particle size ratios⁴⁸. The composition of the nano-alloys influences the phase transition process and therefore their crystal structures and defect formation. The formation

of defects implies stress build-up in individual nanoparticles⁴⁹. When cooled down, the non-eutectic nano-alloys experience gradual and successive growth of different phases and a special liquid-solid transient state coexists depending on their compositions. In contrast for eutectic nano-alloy particles, different phases crystallize simultaneously and more localized during the same cooling condition³⁵. Such rapid yet simultaneous crystallization does not allow stress to dissipate within individual particles. Therefore, more grain boundaries and hence defects can be formed during the solidification of the eutectic nano-alloys.”

Fig. B3 Crystallographic characterization of the $\text{Bi}_x\text{Sn}_{1-x}$ nano-alloys. **a-d** DF-TEM images of individual particle of different $\text{Bi}_x\text{Sn}_{1-x}$ nano-alloy samples. The insets present the FFT patterns of a $40 \text{ nm} \times 40 \text{ nm}$ region of each particle as indicated by the dash-line boxes. **e-h** BF-HR-TEM images of the nano-alloys with their grain boundaries outlined. The insets present their respective grain size distribution. **i-k** BF-HR-TEM images showing extrinsic point defects indicated by arrows (**i**), and line defects (edge dislocations) indicated by T-shape symbols (**j**, **k**). **l** TEM of a eutectic nano-alloy particle and a cartoon shows its heterostructures based on the characterization results.

(4) Eutectic melting of the nano-alloys: The melting characteristics of the eutectic nano-alloy sample is further examined by using the temperature derivative of heat flow as shown in Fig. B4 (corresponds to the inset figures of Fig. 3k of the revised manuscript). The derivatives of heat flows of all non-eutectic samples show temperature-dependent variations (the $x = 0.40$ sample shows a significant slope) while that of the eutectic sample $x = 0.57$ remains constant and flat. The results mean that the phase transition takes place beyond the main peak region for the non-eutectic samples. Therefore, it can be confirmed that only the $x = 0.57$ nano-alloy sample is eutectic. The underlying thermodynamics govern the systems enforce the phase transition between the solid phase ($\text{Bi}_{(s)}$, $\text{Sn}_{(s)}$) and the liquid phase ($\text{BiSn}_{(l)}$) becomes a common feature at the eutectic ratio for different ratios while that of the non-eutectic samples happens successively.

Fig. B4 Plots of heat flows (black curves) and their temperature derivatives (red curves) for different Bi-Sn nano-alloy samples shown in the insets of Fig. 3i of the manuscript.

(5) Ultrasonication and melting of different alloy samples: Ultrasonication process is pursued for creating catalytic nano-alloys as it offers high yield, low cost, and avoid

unwanted by-products that are formed in chemical/electrochemical reactions. It is a non-thermodynamic process and in our study the melting of the bulk samples is achieved by heating at elevated temperatures. Samples made of different Bi-Sn ratios have different melting points and can be processed as liquid metals at different temperature. However, in this study, in order to keep the Bi-Sn ratio as the only variable and keep the synthesis conditions the same for comparison, the same heating temperature was used for processing all samples. For example, temperature changes the surface tension of liquid metals and the final particle size as well, since it is known that the particle size produced by ultrasonication is directly related to surface tension [*Proc. R. Soc. London, Ser. A* 146, 501-523 (1934)]. Discussions in different sections of the manuscript were added in order to address this comment.

Specific comments:

Comment 1: The discussion of heterogeneous catalysis was poor. The authors correctly state that "...evidence shows that bimetallic and multimetallic nano-alloys are superior catalysts relative to their monometallic counterparts^{25,26}." (Line 70), but this does not apply to any arbitrary pair of metals, as the authors imply by this work. They conclude from their TEM work that "The atomic-scale defect containing eutectic Bi-Sn nano alloy, with the highest frequency of point defects, edge and screw dislocations, is therefore expected to be more catalytically active in comparison to its non-eutectic counterparts." (Line 306), which is not a proper assumption either.

Response: Thanks for the comment. We carefully edited the manuscript to assure that a correct conclusion is made.

In our study, we confirm that increasing Bi content ($x=0.20$ to 0.57) within the nano-alloy leads to a decrease in average grain size with the eutectic nanoalloy demonstrating the smallest grain size, indicating that the sample has the highest density of defects (Fig. B3e-h). This increase in defect density correlates well with the CO₂RR activity where the partial current density of formate (j_{HCOO^-}) increases as x is changed from 0.20 to 0.57 . Subsequent increase in x to 0.80 is demonstrated to lead to an increase in grain size indicating less defects and this is mirrored by the decreasing j_{HCOO^-} when compared to eutectic sample. On the basis of these findings, we can conclude that the changing defects in the nanoalloys is dictating its electrocatalytic performance. These defect sites are well reported in literature to improve CO₂RR activity [*Angew. Chem. Int. Ed.* 57, 6054 (2018); *ACS Sustain. Chem. Eng.* 6, 1670

(2018)] as the defect sites govern the adsorption of CO_2 reactants and can also stabilize the formate anion radical intermediates [*Nat. Commun.* 8, 14503 (2017)].

Fig. B3 Crystallographic characterization of the $\text{Bi}_x\text{Sn}_{1-x}$ nano-alloys. **a-d** DF-TEM images of individual particle of different $\text{Bi}_x\text{Sn}_{1-x}$ nano-alloy samples. The insets present the FFT patterns of a 40 nm × 40 nm region of each particle as indicated by the dash-line boxes. **e-h** BF-HR-TEM images of the nano-alloys with their grain boundaries outlined. The insets present their respective grain size distribution. **i-k** BF-HR-TEM images showing

extrinsic point defects indicated by arrows (i), and line defects (edge dislocations) indicated by T-shape symbols (j, k). I TEM of a eutectic nano-alloy particle and a cartoon shows its heterostructures based on the characterization results.

To address the comment, the followings were added to the manuscript:

“Fast Fourier transformation (FFT) leads to more pronounced omnidirectionally dispersed patterns for the eutectic sample, indicating more enhanced spatial misorientation of grains within the eutectic nano-alloys in comparison with the non-eutectic nanoparticles. These results mean that, after solidification, phase separation imposes itself more delicately on the eutectic nano-alloys. As shown in Fig. 5e-h, we can further outline the grain boundaries of the bright-field high-resolution TEM (BF-HR-TEM) images taken from different nano-alloy samples according to various orientations of crystal lattices of different grains. Then the size of the grains within individual nanoparticles can be statistically evaluated. The surveys of the grain size distribution of different samples reveal that the overall grain size of the eutectic sample is the smallest among all the samples (insets of Fig. 5e-h), while its particle size is not (Fig. 3e-h). Crystal grains contain two-dimensional defects. Logically the eutectic sample with the smallest grains should have the largest amount of their boundaries and interfaces. We note that the higher polycrystallinity (smaller grain size) in the eutectic nano-alloys is typically found in the HR-TEM images (Supplementary Fig. 8).

In addition, the formation of low-dimensional defects is frequently observed in the eutectic nano-alloys (Fig. 5i-k), which is either absent or rarely seen in the non-eutectic particles. For instance, the locally distorted lattice structures shown in Fig. 5i are likely induced by the substitution of atom species, especially for the oxides, which can be classified as zero-dimensional point defects. Classic one-dimensional edge dislocations are also identified (Fig. 5j, k) at much higher frequencies of occurrence from the eutectic nano-nanoalloys. It can also be seen from Fig. 5j that due to the small scale of grains and the coexistence of local edge dislocations, their glide planes are terminated shortly. Fig. 5k further shows more complex scenarios in which long-range lattice misalignments occur on both sides of the indicated edge dislocation. Presumably, this is formed as screw dislocations set in and couple with the edge dislocations (mixed dislocations)⁴⁶, or by faulted stacking of different atom species. Based on all the characterizations of the nano-alloys, an illustration of their characteristic structure is presented in Fig. 5l. The nano-alloys have a surface containing SnO and Bi₂O₃, a deeper region mostly composed of SnO and SnO₂, and a Bi-Sn metallic core. All these regions feature intense defects.

Defect formation is known to be facilitated in multi-metal systems due to differences in radius and electron structure of the atoms⁴⁷. The Bi-Sn nano-alloys prepared by liquid phase sonication are expected to have enhanced defect formation since, besides the incorporation of different atom species, their crystal structures are established through a phase transition process³⁵. For the purpose of comparison, we note that nano-alloys grown by other methods usually feature ‘near-perfect’ crystal

structures or large grain size-to-particle size ratios⁴⁸. The composition of the nano-alloys influences the phase transition process and therefore their crystal structures and defect formation. The formation of defects implies stress build-up in individual nanoparticles⁴⁹. When cooled down, the non-eutectic nano-alloys experience gradual and successive growth of different phases and a special liquid-solid transient state coexists depending on their compositions. In contrast for eutectic nano-alloy particles, different phases crystallize simultaneously and more localized during the same cooling condition³⁵. Such rapid yet simultaneous crystallization does not allow stress to dissipate within individual particles. Therefore, more grain boundaries and hence defects can be formed during the solidification of the eutectic nano-alloys.”

And

“The introduction of defects in electrocatalysts are reported to improve CO₂RR^{52,56}. The defect sites govern the adsorption of CO₂ reactants and can also stabilize the formate anion radical intermediates⁵⁷. In addition, grain boundary defects are found to be beneficial for CO₂RR as the breaking of local spatial symmetry near these defect sites tunes the binding energy of the reaction intermediates⁵⁸. Let us recall that the smallest gran sizes are obtained for the eutectic sample (Fig. 5e-h), which results in the augmentation of defects. This correlates well with our experimental findings and explains why the eutectic nano-alloy with the largest amount of grain boundaries, vacancies and dislocations, but not the smallest particle size, offers the highest selectivity and j_{HCOO^-} .”

Comment 2: Fig 2 is not that helpful, since the length scale is much larger than those of the nano-alloys.

Response: Thanks for commenting on the bulk alloy characterizations in Fig. 2. In order to assure that the value of this figure is better presented we carefully edited the paper. We also presented the value of bulk material analysis in “Phase distribution at atomic scale” in response to your previous comment.

The study of the phase separation after solidification of liquid metal alloys in bulk form is a good indication of what may occur in nano-alloys. As shown in our study, the characterizations of bulk alloys can give information about composition (e.g., eutectic/non-eutectic ratio) and structural information (e.g., phase separation in the Bi-Sn system) of the phase separation in samples with no applied ultrasonication. The comparison between bulk

alloys and nano-alloys can reveal similarities and differences across scales, which is useful for fundamentally exploring the influence of liquid phase ultrasonication. We understand that the distribution of different phases in the bulk alloys may not necessarily hold the same trends for the nano-alloys, since some of the governing mechanisms at macroscale and nanoscale are different. However, our results suggest that, at least for the Bi-Sn system, there exists similarity across different scales in the view of phase separation and selective surface oxidation. Given such considerations, we would like to keep Fig. 2 in our manuscript. However, addressing your comments, we carefully revised manuscript to bring out the importance of this figure and established its relevance to the other sections of the paper.

Comment 3: It is difficult to ascertain what structures are the nano-versions of the $\text{Bi}_x\text{Sn}_{1-x}$ compositions. Idealized cartoon schematics would have helped summarize their characterization results in a clearer manner.

Response: Based on all our characterizations, a clear picture of the nano-alloys can be acquired, which have a surface containing Bi_2O_3 and SnO , a deeper region composed of SnO and SnO_2 , and a Bi-Sn metallic core with intense defects (Fig. B5). The figure has also been added to Fig. 5 of the revised manuscript.

Fig. B5 A TEM image of a nano-alloy particle and an illustration shows its heterostructures based on the characterization results.

Comment 4: The choice of reactions was curious. There was no a priori reason for Bi-Sn to be active for CO₂RR. There was no discussion or evaluation of benchmark CO₂RR materials; the authors should comment on how their best material compares to the commonly studied CO₂RR material. As the authors conclude, the photocatalytic degradation of Congo red was due to Bi₂Sn₂O₇ (and SnO₂) being partially formed after calcining their eutectic nano-alloy powder. Bi₂O₃ was also formed, but it is rarely studied as a photocatalyst, in spite of what the authors imply (line 356). The authors did not explain how the material was better, but wrote that "the results suggest that it may be a general trend for the eutectic ratio to be the optimum for developing catalysts using liquid phase ultrasonication." (line 385)

Response: Both Bi and Sn are well-reported electrocatalysts for the conversion of CO₂ (Table B1, adopted as Table S1 in revised Supporting Information) into formate, a by-product that is used in pharmaceuticals and in garments industries. Moreover, recently, alloyed combination of Sn and even Bi-Sn have been investigated for CO₂RR [*Adv. Energy Mater.* 8, 1802427 (2018)]. The addition of Bi to Sn is reported to uplift the electronic state of Sn away from the Fermi level and this allows the favourable adsorption of HCOO⁻ intermediate, allowing improved CO₂RR activity. As a result of these priori work and understanding, we utilized the different Bi-Sn nanoalloys prepared through a liquid metal ultrasonication approach for CO₂RR.

In addition, as shown in Fig. B6b, we present Raman results featuring the intensity of the SnO₂ defect mode at 577 cm⁻¹ and active mode 633 cm⁻¹. It shows that the annealed eutectic sample has the highest relative defect intensity I_D/I_A . We have also compared the compositional dependence of rate constant k and the relative SnO₂ defect intensity I_D/I_A in Fig. B6e. The good correlation between k and I_D/I_A confirms that defect is primarily responsible for the photocatalytic process.

Table B1. CO₂RR catalytic performances of various Sn and Bi-based catalysts.

Catalyst	Electrolyte	Operating Potential (V vs RHE)	Current Density (mA/cm ²)	Faradaic Efficiency for HCOO ⁻ (%)	Reference
Eutectic Bi-Sn	0.1 M KHCO ₃	-1.1	10.7	78	This work
m-SnO ₂	0.1 M KHCO ₃	-1.15	10.8	75.2	1
SnO ₂ /Graphene	0.1M NaHCO ₃	-1.16	10.2	93.6	2

1D SnO ₂ wire in tube	0.1 M KHCO ₃	-0.99	7	63	3
Heat-treated Sn dendrite	0.1 M KHCO ₃	-1.36	17.1	71.6	4
SnO ₂ nanosheets on carbon cloth	0.5 M NaHCO ₃	-0.99	45	87	5
Bi-Sn	0.5 M KHCO ₃ with 600 rpm stirring	-1.14	-61	94	6
Sn/SnOx/Ti	0.5 M NaHCO ₃	-0.7	1.8	55	6
Annealed Sn NPs	0.1M KHCO ₃	-1.2	4	51.5	7
Bi nanosheets	0.5 M NaHCO ₃	-0.8	-5	94	8
Sn quantum sheets/Graphene	0.1 M NaHCO ₃	-1.15	21.5	89	8
BiNS	0.5 M NaHCO ₃	-1.5 V vs SCE	-11	95	11
SnO ₂ nanowires (plasma treated)	0.1 M KHCO ₃	-0.8	6	81	10

References for Table R1

1. Daiyan, R., Lu, X., Saputera, W. H., Ng, Y. H. & Amal, R. Highly selective reduction of CO₂ to formate at low overpotentials achieved by a mesoporous tin oxide electrocatalyst. *ACS Sustain. Chem. Eng.* **6**, 1670-1679 (2018).
2. Zhang, S., Kang, P. & Meyer, T. J. Nanostructured tin catalysts for selective electrochemical reduction of carbon dioxide to formate. *J. Am. Chem. Soc.* **136**, 1734–1737 (2014).
3. Fan, L., Xia, Z., Xu, M., Lu, Y. & Li, Z. 1D SnO₂ with wire-in-tube architectures for highly selective electrochemical reduction of CO₂ to C₁ products. *Adv. Funct. Mater.* **28**, 1–8 (2018).
4. Won, D. H. *et al.* Rational design of a hierarchical tin dendrite electrode for efficient electrochemical reduction of CO₂. *ChemSusChem* **8**, 3092–3098 (2015).
5. Li, F., Chen, L., Knowles, G. P., MacFarlane, D. R. & Zhang, J. Hierarchical mesoporous SnO₂ nanosheets on carbon cloth: a robust and flexible electrocatalyst for CO₂ reduction with high efficiency and selectivity. *Angew. Chem. Int. Ed.* **56**, 505–509 (2017).
6. Chen, Y. & Kanan, M. W. Tin oxide dependence of the CO₂ reduction efficiency on tin electrodes and enhanced activity for tin/tin oxide thin-film catalysts. *J. Am. Chem. Soc.* **134**, 1986–1989 (2012).
7. Wu, J., Risalvato, F. G., Ma, S. & Zhou, X.-D. Electrochemical reduction of carbon dioxide III. The role of oxide layer thickness on the performance of Sn electrode in a full electrochemical cell. *J. Mater. Chem. A* **2**, 1647 (2014).
8. Lei, F. *et al.* Metallic tin quantum sheets confined in graphene toward high-efficiency carbon dioxide electroreduction. *Nat. Commun.* **7**, 12697 (2016).
9. Fu, Y. *et al.* Novel hierarchical SnO₂ microsphere catalyst coated on gas diffusion electrode for enhancing energy efficiency of CO₂ reduction to formate fuel. *Appl. Energy* **175**, 536–544 (2016).
10. Kumar, B. *et al.* Reduced SnO₂ porous nanowires with a high density of grain boundaries as catalysts for efficient electrochemical CO₂-into-HCOOH conversion. *Angew. Chem. Int. Ed.* **56**, 3645–3649 (2017).

Fig. B6 Characterization and photocatalytic activity of the annealed $\text{Bi}_x\text{Sn}_{1-x}$ nano-alloys. **a** XRD patterns of the eutectic sample after annealing at $500\text{ }^\circ\text{C}$ for 1 hr. The inset shows the colour of the eutectic sample before and after annealing. **b** Raman spectra of the $\text{Bi}_x\text{Sn}_{1-x}$ nano-alloys annealed at $500\text{ }^\circ\text{C}$ for 1 hr. The magnified regions show the relative intensity of the SnO_2 defect mode I_D (577 cm^{-1}) and active mode I_A (633 cm^{-1}). **c** Plots of $\ln(c_0/c_t)$ vs t for the eutectic samples annealed at different temperatures. **d** Plots of $\ln(c_0/c_t)$ vs t for $\text{Bi}_x\text{Sn}_{1-x}$ nano-alloy samples annealed at different $500\text{ }^\circ\text{C}$ for 1 hr. **e** The dependence of SnO_2 defect intensity (characterised by I_D/I_A) and dye degradation rate constant k on the mixing ratio of the $\text{Bi}_x\text{Sn}_{1-x}$ nano-alloy samples.

Comment 5: Line 334: what is "cocking"?

Response: The term "coking" commonly refers to the blocking of active sites of catalysts through van der Waals adhesion of carbonaceous species formed during catalysis process itself. Using liquid metal catalysts eliminated the deactivation via coking since the liquid metal surface removes van der Waals forces [*Nat. Chem.* 9, 862 (2017)].

Comment 6: The experimental design for this work was incomplete. Control samples were not tested, for example, pure SnO₂, pure Bi₂O₃, conventionally prepared bimetallic materials, benchmark catalysts.

Response: Thanks for the comment. To address your comment, we further prepared control samples of Bi and Sn for CO₂RR and annealed them for dye degradation experiments.

As shown in Fig. B7a, b, the TEM images and AFM results show that when either liquid Sn or liquid Bi is used, the ultrasonication process mostly produces nanoplates rather than spherical nanoparticles (spherical Sn nanoparticles are also seen occasionally). As can be seen from the AFM, the Sn and Bi nanoplates typically have a few tens of nanometre thickness, while they are a few hundred nanometres in other dimensions. This is due to the formation of two-dimensional oxide of these metals during the sonication that can be naturally delaminated from the surface of the liquid. This shows one of the stark differences between the alloys and pure metals. It seems that the layering of the surface which is seen for pure Bi and Sn in liquid state [*Phys. Rev. B* 55, 15874 (1997)] promotes the formation of the flakes. The layering is not favoured in the Bi-Sn liquid alloy and the coexistence of Bi and In in the liquid alloys is not favourable for the metals to separate themselves from the bulk as flakes during the sonication.

Further characterizations by DSC (Fig. B7c), XRD (Fig. B7d), XPS (Fig. B7e), and Raman (Fig. B7e, the Raman spectra of the annealed Sn and Bi samples are Fig. B6b) showed the consistent results. One very important observation from these experiments is the difference between the surface oxidation of the bulk and oxides on nano-alloy surface. This deviation is presumably caused by the sonication process that promotes particular phase separations at the same time as the surface oxidation takes place. Sonication applies energy to the system and as such can cause phase separation which allows the emergence of Bi₂O₃. Additionally, the high energy applied via sonication allows the higher oxidation stoichiometry of SnO₂ occurs. The results for electrochemical CO₂ reduction and the dye degradation experiments are shown in Fig. B8 and Fig. B9, respectively. In general, the dye degradation rate constant and the formate selectivity of the control samples is lower in comparison to the Bi-Sn nano-alloys.

Fig. B7 **a, b** Control samples prepared with Sn metal (**a**) and Bi metal (**b**). The inset figures show the AFM topography and the thickness profile along the dash line of Sn and Bi particles. **c** DSC, **d** XRD, **e** XPS, and **f** Raman results of the control samples.

Fig. B8 Dependence of Faradaic efficiency for HCOO^- on applied potential for eutectic BiSn nanoalloy and control Sn and Bi electrocatalysts.

Fig. B9 Plots of $\ln(c_0/c_i)$ vs t for the control Sn and Bi photocatalysts.

Comment 7: Figure 7; Line 490: "The degradation rate of the dye was studied" but only degradation % was reported. Properly, the rate constants should have been reported and compared. Percent loss of Congo red is important observe but does not fully and quantitatively capture the differences in catalyst performance.

Response: Thanks for your helpful advice. We have redrawn the results for dye degradation and plotted $\ln(c_0/c_i)$ vs t in Fig. B6c, d. The linear curves indicate the samples follow a pseudo-first order degradation kinetics towards the degradation of the Congo red as the model dye. Now the activity of different samples can be directly compared by looking at the slope of the $\ln(c_0/c_i) - t$ curves (the rate constant k), showing the optimal annealing temperature (500 °C) in Fig. B6c and optimal mixing ratio (eutectic ratio, $x = 0.15$) in Fig. B6d, respectively.

Responses to Reviewer #3:

Comment: First of all, well written abstract and introduction. If the story can go as described in the abstract, and the assumption made in the introduction can be validated, there is no doubt that this is a Nat Comm level work.

However, after going through the full text, the reviewer is quite disappointed indeed, failing to see what are promised in the abstract and introduction turn into reality. The reviewer does not believe the authors have a convincing story here and is also challenging the authors' knowledge on eutectic alloys for them to be qualified enough to write a story based on eutectic alloys.

Response: Firstly, we would like to thank you for showing your interests in our work and your help and time that you spent on reviewing our manuscript. We carefully read your comments and we edited the whole manuscript and added a large number of extra experiments to assure that your comments have been fully addressed. We edited the layout of the manuscript to assure that necessary explanations and clarifications are incorporated. Based on your comments and the comments from the other reviewers, we have modified the manuscript, significantly enhanced its scientific contents and included more in-depth discussions to support the conclusions. The manuscript has been improved considerably. Please find our point-to-point responses to your specific comments. We hope that, with the improvements been made according to your suggestions, and our revised manuscript now meet with your approval for publication.

Comment 1: They talk too much, almost unnecessarily, about the basics of eutectic alloys, and use the wording like “Strikingly, when the Bi-Sn ratio reaches the eutectic value ($x = 0.57$), the alternately arranged Bi and Sn lamellae become dominant, ruling out the discrete growth regime as observed in the hypoeutectic samples.” Well, this is nothing striking at all to people knowing what eutectic alloys mean. This also relates to the description on the DSC behaviours. Nevertheless, these statements are basically unnecessary, but not wrong. They then start to go wrong afterwards, starting from using “split of the DSC peaks” to describe what is seen in Fig.3, then to “shoulders of the non-eutectic samples are shaved after the liquid phase ultrasonication, indicating accompanying compositional changes”, and finally to “the single melting peak of the eutectic sample implies that the eutectic Bi-Sn ratio is still maintained after ultrasonication”. At this point, the reviewer already lost the interest to

continue the reading. Apparently, they do not know much about eutectic alloys, and it seems that they also do not know much about thermal measurements.

Response: Thanks for pointing out the phrasing problem. We carefully edited the whole paper and removed these subjective terms to assure your concerns are met.

Regarding the DSC measurement of the nanoparticles, this unusual observation lead to the conclusion that likely two sub-particle types co-exist with different compositions in the samples. Such a conclusion is also confirmed by observations in Fig. C1 and Fig. C2b, 2c below. Peculiar phase separation occurs during sonication that has also been shown by Tang *et al* [Tang, S.-Y. *et al.* Matter (2019)]. Based on our TEM-EDX mapping results, the peaks below 250°C and above 250°C are assigned to the small particles (Fig. C2b) and the big particles (Bi-rich, Fig. C2c), respectively. Therefore, the two major peaks in DSC curves represent the melting of different types of particles. Since the small particles account for the majority of the surface of the samples and will dominant in catalysis reactions, our discussions mainly focus on them. Therefore, the peak below 250°C, which has a shoulder on its right side (not the melting peak at >250°C), is discussed in detail.

Eutectic system is defined by its single-temperature phase transition behaviours. As indicated in the inset of Fig. C2a, only the eutectic nano-alloy sample (the small particles) shows a single melting peak. The shoulders of the melting peaks for the two samples $x = 0.20$ and $x = 0.80$ are obvious and the sample $x = 0.40$ sample show a smooth transition. However, the $x = 0.40$ sample should be non-eutectic since its melting transition covers a wide temperature range from about 145°C to 220°C, so the melting of the $x = 0.40$ sample takes places at temperature far beyond the main peak. In addition, this can also be illustrated using the temperature derivative of heat flow as shown in Fig. C3. The temperature derivatives of heat flows of all non-eutectic samples show temperature-dependent variations (the $x = 0.40$ sample shows a slope) while that of the eutectic sample $x = 0.57$ remains constant. The results mean that the phase transition takes place beyond the main peak region for all three non-eutectic samples. Therefore, it can be confirmed that only the $x = 0.57$ nano-alloy sample is eutectic. Fig. C3 has been added to the revised Supporting Information. In addition, we revised this section in the manuscript to make it clearer to the readers.

Fig. C1 TEM images of the $\text{Bi}_x\text{Sn}_{1-x}$ nano-alloy samples prepared by the liquid-phase ultrasonication method. Occasionally, some particles much larger than the average dimensions are also seen.

Fig. C2 a DSC curves of the samples prepared by ultrasonication. b EDX mapping of small particles. c EDS mapping of large particles.

Fig. C3 Plots of heat flow (black curve) and its temperature derivative (red curve) for the different Bi-Sn nanoalloy samples shown in the insets of Fig. 3i of the manuscript.

Comment 2: They made such a strange statement in Page 9: This indicates that under the current liquid phase ultrasonication conditions, the particle size is not determined by the melting point of the bulk samples, instead, it correlates with the Bi-Sn mixing ratio. So for them, the melting point does not change with the Bi-Sn mixing ratio? The reviewer even wants to challenge their knowledge on thermodynamics now.

Response: Thanks for the comment. We believe that there is a misunderstanding that perhaps came from lack of clarity in the text. To address this issue, we edited some of the sentences to assure that there would be no confusion.

The Bi-Sn mixing ratio affects the melting point of the samples. Actually, the composition dependence of melting point is one of the starting points of our study and we take advantage of the melting point drop to process the alloys as liquid. We wished to express that the main variant that influences the particle size during the ultrasonication process is surface tension which is impacted by the Bi-Sn ratio. The dynamics governing the particle size of ultrasonication is depicted by Taylor's formula as:

$$d = 2\sigma/(\eta_c\dot{\gamma}) \quad (1)$$

where σ is the interfacial tension that shares the same composition dependence as the Bi-Sn alloy surface tension, η_c is the viscosity of the solvent, and $\dot{\gamma}$ is the shear rate of the flow produced by ultrasonication [*Proc. R. Soc. London, Ser. A* 146, 501-523 (1934)]. Since the increase of Bi ratio reduces the surface tension of the Bi-Sn alloys [*J. Electron. Mater.* 30, 1104-1111(2001)], it matches our results that the overall size of the particles decreases as Bi ratio increases. This explains why the eutectic sample which has the lowest melting temperature does not generate the smallest particles.

The Bi-Sn ratio influences many properties of the alloys (including melting point). In our study, in order to keep the Bi-Sn ratio as the only variable and keep the synthesis conditions the same for comparison, the same processing temperature was used. Therefore, we stated that the size of the obtained particles is not determined by the melting point of the bulk samples. We revised the part carefully to avoid misunderstandings by the readers.

Comment 3: In Page 12, quite annoyingly, they started to describe the co-existence of Bi oxide, immediately after they just explained why Bi oxide is absent.

Response: Thanks for the comment. We understand the confusion as the text was not clear. We carefully edited to assure that a correct conclusion is reached. We have also adjusted Fig. 3 and Fig. 5 (see Fig. C4 and Fig. C5 below) so that the Raman results from bulk samples and particle samples are presented separately for clarity. A Bi-Sn-O phase diagram is also added to confirm our discussions about the selective oxidation of the system. This part of the text has been changed as follows:

“The XRD technique, which has deep penetration, allows a glimpse of the inner particle compositions of the particle samples. More importantly, the outcome will tell whether the knowledge from the XRD results of the bulk alloy can be applied to the core of nano-alloys. Encouragingly as shown in Fig. 3l, the XRD patterns reveal that the Bi and Sn metallic phases in the particles share the same crystallographic phases with their bulk precursors and the samples are partially oxidized after ultrasonication. SnO and Bi₂O₃ are recognized as the oxide phase for the control Sn and Bi samples, respectively. For the Bi-Sn alloy particles, two types of Sn oxides, namely SnO and SnO₂, emerge, but the oxide of Bi is absent. The absence of Bi oxide deep inside the nano alloy particles can be rationalized by considering the competing oxidizing process of the metallic phases based on the Bi-Sn-O phase diagram. The oxidation mechanisms of pure Sn and Bi is likely due to the constant exfoliation of the flakes during the sonication due to surface layering, which is also similarly seen in the sonication of Ga [*Adv. Funct. Mater.* 27, 1702295 (2017)]. The difference in morphologies between the pure and alloyed samples shows the significant impact of alloying, which increases the surface entropy and disturbs its order, reducing the possibility of removing surface layers by shear force during the sonication process.

The surface of the particles may constitute different compositions from their core. Understanding the surface composition is especially important for nano-alloys as oxides emerge on or within nanoparticle, while they are not seen in the bulk. The coexistence of Bi and Sn oxides on the surface of the Bi-Sn nano-alloy samples can be confirmed further by EDX-coupled scanning TEM (STEM) and X-ray photoelectron spectroscopy (XPS), which are both surface sensitive techniques. STEM-EDX mapping shows the distribution of Bi and Sn, together with O due to partial oxidation (Fig. 4a and Supplementary Fig. 4-6). The XPS twin peaks at the Bi 4f region (159.4 and 164.6 eV) and the Sn 3d region (486.7 and 495.2 eV) are characteristic features of Bi₂O₃, and SnO/SnO₂, respectively (Fig. 4b). From the variation of the intensity of the Bi 4f and Sn 3d peaks, it can be inferred that the alloys’ compositional influences are peculiarly expressed on the surface of the nano-alloys.”

Fig. C4 Characterization of $\text{Bi}_x\text{Sn}_{1-x}$ bulk alloys. **a** SEM images showing the microscopic morphologies of the transverse sections of the bulk samples with different Bi ratios as indicated. **b-e** EDX element mapping showing the distribution of the Bi and Sn phases in four types of microstructures. Regions featuring different microstructure types found in different samples in **a** are outlined by boxes with the corresponding colours. **f**, DSC curves showing the melting (solid lines) and solidification (dash lines) trends of the bulk samples. **g**, XRD patterns of the bulk samples. **h** Schematics of touch-printing surface oxide layer from liquid $\text{Bi}_x\text{Sn}_{1-x}$ bulk samples. **i** Raman spectra of the touch-printed surface oxide layers from bulk liquid $\text{Bi}_x\text{Sn}_{1-x}$ samples. **j** Phase diagram of the Bi-Sn-O system calculated at 300 °C, 1 atm.

Fig. C5 Surface composition analysis of the $\text{Bi}_x\text{Sn}_{1-x}$ nano-alloys. **a** TEM image and STEM-EDX element mapping, showing the distribution of Bi, Sn and O in a nano-alloy particle made of the eutectic Bi-Sn. **b** XPS spectra of different $\text{Bi}_x\text{Sn}_{1-x}$ nano-alloys. The same scale is used for the XPS intensity for different elements of the same sample unless otherwise specified in the figures. The significant difference of the relative intensity of O1s peaks between the nano-alloy samples and the control Bi and Sn samples may come from the different oxygen dissolvability of solvent that used to prepare the samples (see Method). **c** Raman spectra of different $\text{Bi}_x\text{Sn}_{1-x}$ nano-alloys.

A comprehensive discussion was also added to the body of the manuscript as follows:

“In the Bi-Sn binary alloy system, the oxidation of the liquid metal surface competes between the formation of SnO, SnO₂, and Bi₂O₃. This is assumed to be governed by the Gibb’s free energy of oxidation when no sonication is applied¹⁵. To assess this assumption, we exfoliate the surface oxide layers formed on different liquid $\text{Bi}_x\text{Sn}_{1-x}$ bulk samples to examine which oxide phase is selectively formed with no externally applied energy (Fig. 2h and Supplementary Fig. 2). The Raman spectra of the surface oxide layers (Fig. 2i) show that the result of the competition of oxide formation on the surface of bulk Bi-Sn alloy is preferentially won by SnO. Here Bi₂O₃ layer is rarely seen, even at high concentrations of Bi, and SnO₂ only shows itself at $x = 0.80$. The favourability of SnO formation on the surface of bulk is also validated through thermodynamic calculation using FACT-Sage³⁶. As can be seen from Fig. 2j, the formation of SnO can be predicted from the Bi-Sn-O phase diagram at low O₂ concentration (region i) which is matched by our experimental conditions. In addition, the formation of SnO₂ at low Sn ratio (regions ii and iii) in the system can be inferred from the phase diagram.”

“One important observation from these experiments is the difference between the surface oxidation of the bulk and oxides on nano-alloy surface. This deviation is presumably caused by the sonication process which promotes phase separation at the same time as the surface oxidation takes place. We have already discussed about the surface domination of the bulk sample by SnO when no sonication is applied. However, our characterizations revealed that under ultrasonication SnO₂ also appears near the surface when Sn concentration reaches the eutectic value or smaller. For these concentrations the emergence of SnO₂ is favoured, especially when oxygen can be continuously dissolved in the environment, which is the case for the sonication process. Sonication also applies energy to the system and as such can cause phase separation, which allows the emergence of Bi₂O₃.”

Comment 4: Most importantly, the reviewer does not see eutectic nano-alloys are much superior to non-eutectic nano-alloys, regarding the catalytic and photocatalytic activity, from what they show in Fig. 6 and Fig. 7, and also does not see the claimed evidence for this so-called superiority, which is the enhanced defects. This is the kernel of this story, and it is rather weakly supported.

Response: Thanks for the comment and we understand your concern. To address this comment, we added significant number of measurements and assessment to strengthen the conclusions of the paper.

We made substantial changes to the body of the manuscript and Supplementary Information. We modified the TEM characterization part to support our discussions on defect formation. We obtained more HRTEM images for different Bi-Sn nano-alloy samples. According to different orientations of the lattice fringes, we outlined the grain boundaries, based on which we were able to evaluate the grain size of the nano-alloys. In doing so, we measured both the long axis and short axis of irregular-shape grains and their average value was used to present the characteristic grain size. The grain size distributions of different samples were included in the inset of Fig. C6e-h below (Fig. 5e-h of the revised manuscript) and they indeed show that the eutectic sample has the smallest grain size in comparison with the non-eutectic ones. Grain boundaries are defects which are beneficial for catalytic process [*J. Am. Chem. Soc.* 137, 4606-4609 (2015)]. For example, the binding energy of the reaction intermediates in CO₂ reduction reaction can be tuned near these defect sites due to local spatial symmetry breaking [*Angew. Chem. Int. Ed.* 56, 3645-3649 (2017)]. Therefore, the eutectic nano-alloy samples with the smallest grain size (and not the particle size) can provide

more defects sites and demonstrates the highest selectivity and current density among all the samples during the CO₂ reduction reaction.

We also reworked the results and figures for dye degradation and plotted $\ln (c_0/c_t)$ vs t in Fig. C7c, d. The linear curves indicate the samples follow a pseudo-first order degradation kinetics towards the degradation of the Congo red model dye, governed by the relation:

$$-\frac{dc}{dt} = kc \quad (2)$$

where c is the concentration of dye and k the observed rate constant (slope of the $\ln (c_0/c_t) - t$ curves). The activity of different samples can be directly compared by looking at the slope of the $\ln (c_0/c_t) - t$ curves (the rate constant k), showing the optimal annealing temperature (500 °C) in Fig. C7c and optimal mixing ratio (eutectic ratio, $x = 0.15$) in Fig. C7d, respectively.

Additionally, in the new version of the paper, it is discussed that SnO₂ has a defect Raman mode I_D at 577 cm⁻¹ and a nearby active Raman mode I_A at 633 cm⁻¹ [*J. Phys. Chem. C* 115, 118-124 (2010)]. We also compared the compositional dependence of rate constant k and the relative SnO₂ defect intensity I_D/I_A in Fig. C7e. The good correlation between k and I_D/I_A confirms that defect is primarily responsible for the photocatalytic process.

As we can be seen, both experiments on catalysis (electrochemical and photocatalysis) demonstrate eutectic sample has the highest activity. Moreover, both different defect characterizations methods reveal the eutectic sample has more defects. Therefore, we attribute the better performance of the eutectic sample to its enhanced defect formation.

The following Fig. C6e-6i, Fig. C6l, Fig. C7e were also added to the paper to provide the further evidence.

Fig. C6 Crystallographic characterization of the $\text{Bi}_x\text{Sn}_{1-x}$ nano-alloys. **a-d** DF-TEM images of individual particle of different $\text{Bi}_x\text{Sn}_{1-x}$ nano-alloy samples. The insets present the FFT patterns of a $40 \text{ nm} \times 40 \text{ nm}$ region of each particle as indicated by the dash-line boxes. **e-h** BF-HR-TEM images of the nano-alloys with their grain boundaries outlined. The insets present their respective grain size distribution. **i-k** BF-HR-TEM images showing extrinsic point defects indicated by arrows (**i**), and line defects (edge dislocations) indicated by T-shape symbols (**j**, **k**). **l** TEM of a eutectic nano-alloy particle and a cartoon shows its heterostructures based on the characterization results.

Fig. C7 Characterization and photocatalytic activity of the annealed $\text{Bi}_x\text{Sn}_{1-x}$ nano-alloys. **a** XRD patterns of the eutectic sample after annealing at 500°C for 1 hr. The inset shows the colour of the eutectic sample before and after annealing. **b** Raman spectra of the $\text{Bi}_x\text{Sn}_{1-x}$ nano-alloys annealed at 500°C for 1 hr. The magnified regions show the relative intensity of the SnO_2 defect mode I_D (577 cm^{-1}) and active mode I_A (633 cm^{-1}). **c** Plots of $\ln(c_0/c_i)$ vs t for the eutectic samples annealed at different temperatures. **d** Plots of $\ln(c_0/c_i)$ vs t for $\text{Bi}_x\text{Sn}_{1-x}$ nano-alloy samples annealed at different 500°C for 1 hr. **e** The dependence of SnO_2 defect intensity (characterised by I_D/I_A) and dye degradation rate constant k on the mixing ratio of the $\text{Bi}_x\text{Sn}_{1-x}$ nano-alloy samples.

The following parts were added to the paper:

“Fast Fourier transformation (FFT) leads to more pronounced omnidirectionally dispersed patterns for the eutectic sample, indicating more enhanced spatial misorientation of grains within the eutectic nano-alloys in comparison with the non-eutectic nanoparticles. These results mean that, after solidification, phase separation imposes itself more delicately on the eutectic nano-alloys. As shown in Fig. 5e-h, we can further outline the grain boundaries of the bright-field high-resolution TEM (BF-HR-TEM) images taken from different nano-alloy samples according to various orientations of crystal lattices of different grains. Then the size of the grains within individual nanoparticles can be statistically evaluated. The surveys of the grain size distribution of different samples reveal that the

overall grain size of the eutectic sample is the smallest among all the samples (insets of Fig. 5e-h), while its particle size is not (Fig. 3e-h). Crystal grains contain two-dimensional defects. Logically the eutectic sample with the smallest grains should have the largest amount of their boundaries and interfaces. We note that the higher polycrystallinity (smaller grain size) in the eutectic nano-alloys is typically found in the HR-TEM images (Supplementary Fig. 8).

In addition, the formation of low-dimensional defects is frequently observed in the eutectic nano-alloys (Fig. 5i-k), which is either absent or rarely seen in the non-eutectic particles. For instance, the locally distorted lattice structures shown in Fig. 5i are likely induced by the substitution of atom species, especially for the oxides, which can be classified as zero-dimensional point defects. Classic one-dimensional edge dislocations are also identified (Fig. 5j, k) at much higher frequencies of occurrence from the eutectic nano-nanoalloys. It can also be seen from Fig. 5j that due to the small scale of grains and the coexistence of local edge dislocations, their glide planes are terminated shortly. Fig. 5k further shows more complex scenarios in which long-range lattice misalignments occur on both sides of the indicated edge dislocation. Presumably, this is formed as screw dislocations set in and couple with the edge dislocations (mixed dislocations)⁴⁶, or by faulted stacking of different atom species. Based on all the characterizations of the nano-alloys, an illustration of their characteristic structure is presented in Fig. 5l. The nano-alloys have a surface containing SnO and Bi₂O₃, a deeper region mostly composed of SnO and SnO₂, and a Bi-Sn metallic core. All these regions feature intense defects.

Defect formation is known to be facilitated in multi-metal systems due to differences in radius and electron structure of the atoms⁴⁷. The Bi-Sn nano-alloys prepared by liquid phase sonication are expected to have enhanced defect formation since, besides the incorporation of different atom species, their crystal structures are established through a phase transition process³⁵. For the purpose of comparison, we note that nano-alloys grown by other methods usually feature ‘near-perfect’ crystal structures or large grain size-to-particle size ratios⁴⁸. The composition of the nano-alloys influences the phase transition process and therefore their crystal structures and defect formation. The formation of defects implies stress build-up in individual nanoparticles⁴⁹. When cooled down, the non-eutectic nano-alloys experience gradual and successive growth of different phases and a special liquid-solid transient state coexists depending on their compositions. In contrast for eutectic nano-alloy particles, different phases crystallize simultaneously and more localized during the same cooling condition³⁵. Such rapid yet simultaneous crystallization does not allow stress to dissipate within individual particles. Therefore, more grain boundaries and hence defects can be formed during the solidification of the eutectic nano-alloys.”

“As shown in Fig. 7b, when compared to the Raman spectra of the control Sn and Bi samples, the predominance of Bi₂O₃^{44,62} and SnO₂^{43,63} in the annealed nano-alloy samples is further confirmed. We highlight that Raman results reveal that the annealed eutectic sample has the highest relative

intensity of SnO₂ defects which is characterized as I_D/I_A , where I_D (577 cm⁻¹) and I_A (633 cm⁻¹) is the SnO₂ defect and active Raman mode, respectively⁶⁴. The results proved another evidence to support our crystallographic defect characterizations that the eutectic nano-alloy sample has more enhanced defects formation than the non-eutectics.

As shown in Fig. 7c and Fig. 7d, the plots of $\ln(c_0/c_t)$ vs t indicate that the annealed nano-alloys demonstrate a pseudo-first order degradation kinetics towards the Congo red model dye, following the relation:

$$-\frac{dc}{dt} = kc \quad (2)$$

where c is the concentration of dye and k the observed rate constant (slope of the $\ln(c_0/c_t) - t$ curves). As shown in Fig. 7c, tests on different annealing temperatures reveal that the eutectic sample offers the highest degradation when annealed at 500 °C, and this annealing temperature is chosen for other samples. Comparison between different sample ratios again shows that the annealed eutectic sample has the largest rate constant k , indicating the highest dye degradation rate (Fig. 7d). In addition, the nano-alloy samples demonstrate higher activity than the control Sn ($x = 0.00$) and Bi ($x = 1.00$) samples (Supplementary Fig. 12). As shown in Fig. 7e, the good correlation between the defect intensity (I_D/I_A) and rate constant k confirms that defects are primarily responsible for the photocatalytic process.”

Comment 5: Page 20: The generation of Bi₂Sn₂O₇, also a visible-light-driven photocatalyst, indicates the fine mixing of Bi and Sn. The reviewer sees no such a connection that can be made here.

Response: Thanks for the comment. In the revised manuscript, we carefully edited the discussion to make the connection. The following was added to the paper:

“Additionally, a binary oxide phase, Bi₂Sn₂O₇ that is generally recognised as a visible-light-driven photocatalyst, is also generated, which means that our method can access more complex catalytic structures.”

Reviewers' comments:

Reviewer #1 (Remarks to the Author):

The authors have made some progress in the revised manuscript by adding some experiments and explanations. However, I think the authors should think about what is the most important point of this paper. The title of this paper was focused on "eutectic alloys", but there were a great portion of writing about the catalytic activity of Bi and Sn oxides. It is confusing that which part the authors want to highlight, the alloys or the oxides? Besides, despite a lot of added discussion about how to define the number of defects in nano-alloys, there is still lack of in-depth discussion on the differences on the mechanisms of catalytic activity among these nano-alloys. The authors cited many references to demonstrate defects play an important role in offering active sites, however, the synergistic effect between Bi and Sn, the influence of different defect types, and so on, are also very possible and should be carefully ruled out. In addition, the contrast in HRTEM cannot be quantified directly. So, "the zero-dimensional point defect" as the authors mentioned so many times in the revised manuscript are wrong. If the authors want to identify point defect, HRTEM image simulation verification or spherical-aberration corrected STEM may help. Also, it is very tricky to use HRTEM to identify the number of grain boundaries and grain size. It is suggested to use other characterization techniques such as EBSD. There is no discussion about grain boundary formation which may be related to different compositions and phase transition thermodynamics.

Reviewer #2 (Remarks to the Author):

The authors' revised manuscript significantly addresses this reviewer's major comments. It was good that the authors acknowledged they are working with a 3-component system (with oxygen being the third one). There are some excellent materials characterization results. The in-depth improvements do highlight additional weaknesses, however.

The authors make several assertions, that are not strongly justified or that are based on assumptions. In writing "...sonicating the Bi-Sn liquid alloys at a mildly elevated temperature..." the authors assert that the bulk material was melted, and that the temperature was higher than room temperature. Elsewhere they write with similar assertions like "The Bi-Sn nano-alloys prepared by liquid phase sonication..." There is no proof that the metals are in the liquified state.

They write, "Peculiar phase separation occurs during sonication that has also been shown by Tang et al.³⁸. These results imply that when the liquid bulk alloys are fragmented into nanodroplets during ultrasonication, the localised composition may be changed."

The solid samples are phase-separated for most compositions. This may be the simpler reason for the observed heterogeneity in the formed nano-particles.

The ultrasonication experiments look too difficult and dangerous for other researchers to replicate. They use a commercial probe sonicator for 2-hr experiments with 10-second on/off cycles. No temperature was recorded or reported. The hot plate apparently can be set to 400C, and they observe some DMSO evaporating during process. DMSO has bp of ~190C, indicating temperatures can reach 190C. The DMSO (and silicone oil?) vapor poses flammability and safety issues.

They cite something called "Taylor's formula," but do not make use of it. I've never heard of this equation before. The reference that they cite is the original 1934 paper by the famous Geoffrey I. Taylor, in which the capillary number (an important equation) is defined.

Figure 2h showing "touch printing surface oxide layer" is not discussed well in text. Touch printing was not defined or elaborated upon.

Lines 385-391. The writing about the coking observation is circuitous and almost nonsensical, to the casual reader. First, carbonaceous deposits should not be called "coke". Coke is specifically formed at high-temperatures (>500C) in gas-phase reactions., almost graphitic in nature. Second, they write that " this is a discussion relevant to above the 139C operating (melting) temperature for which the electrolyte should be replaced." Why would the CO₂ electrocatalytic reduction reaction be operated above 139C?

The authors make the simplistic argument that more defects is better for higher catalytic performance. In catalysis, the choice of metals is critical for activity, because on the affinity of the metal (characterized by its d-band energy) with CO₂. The authors did not explain in manuscript why Bi-Sn nano-alloy catalysts would be active for CO₂ electrocatalytic reduction.

Line 419. The photocatalytic results are not impressive, to this reviewer. There are plenty of properly performed photocatalytic reactions published in the literature (environmental science/engineering journals, for example), that can serve as models for how to carry out the reaction rigorously. The annealed nano-alloys have two phase Bi₂O₃ and SnO₂, as they claim. They show data that a binary oxide phase Bi₂Sn₂O₇ is found in the eutectic material – is this phase also found in the other compositions? Could the different amounts of this photoactive phase Bi₂Sn₂O₇ be the reason for the maximal activity for the eutectic material? The authors did not offer an explanation, but re-stated their observation as: "it may be a general trend for the eutectic ratio to be the optimum for developing catalysts using liquid phase ultrasonication." This was raised in my original comments. The authors gave a more detailed response in the rebuttal document, but none of this was introduced in the revised manuscript.

Reviewer #3 (Remarks to the Author):

The revised version reads more correct at least, by removing previously inappropriate or even wrong statements.

The reviewer has now some new concerns on the revised manuscript.

1. On equation 1 where they tried to defend why the particle size decreases as the Bi ratio increases: they were assuming the surface tension dominates, rather than the viscosity and shear rate. This has to be justified, as otherwise the equation does not mean much.
2. In the end, the reviewer is still confused on what happens indeed during the ultra-sonication process: so a phase separation in the liquid occurs first, and then one separated liquid phase still goes through the eutectic reaction? What drives the phase separation in the liquid state? What is the solid evidence to prove the eutectic reaction indeed occurs, other than what is shown in Fig. 3k, which is certainly not a convincing one? Note that if the liquid phase separation does happens, they cannot assume the eutectic reaction can still necessarily occur, as the composition already shifts.
3. So the eutectic nanoalloy does not have the smallest particles size, but the smallest grain size? This is not convincing on the one hand, based simply from some TEM analysis, and difficult to be perceived on the other hand. Why is that? And more defects in the eutectic nanoalloy? Honestly, the reviewer cannot see a convincing trend here. In the end, the reviewer tends to think being eutectic or not is not the key here. The low melting point for these near-eutectic alloys is the key, which enables the ultra-sonication process. There is probably no much point to emphasize that the eutectic alloy is the magic composition. They cannot prove their nanoalloy is a eutectic nanoalloy, and they also cannot convincingly show the eutectic nanoalloy, if it is eutectic indeed, is much superior to those none-eutectic nanoalloys in terms of catalytic or photocatalytic performance.

Reviewer #1 (Remarks to the Author):

Comment 1: The authors have made some progress in the revised manuscript by adding some experiments and explanations.

Response 1: Thank you very much for your comments and suggestions which enable us to improve our work.

Comment 2: I think the authors should think about what is the most important point of this paper. The title of this paper was focused on “eutectic alloys”, but there were a great portion of writing about the catalytic activity of Bi and Sn oxides. It is confusing that which part the authors want to highlight, the alloys or the oxides? Besides, despite a lot of added discussion about how to define the number of defects in nano-alloys, there is still lack of in-depth discussion on the differences on the mechanisms of catalytic activity among these nano-alloys. The authors cited many references to demonstrate defects play an important role in offering active sites, however, the synergistic effect between Bi and Sn, the influence of different defect types, and so on, are also very possible and should be carefully ruled out. In addition, the contrast in HRTEM cannot be quantified directly. So, “the zero-dimensional point defect” as the authors mentioned so many times in the revised manuscript are wrong. If the authors want to identify point defect, HRTEM image simulation verification or spherical-aberration corrected STEM may help. Also, it is very tricky to use HRTEM to identify the number of grain boundaries and grain size. It is suggested to use other characterization techniques such as EBSD. There is no discussion about grain boundary formation which may be related to different compositions and phase transition thermodynamics.

Response 2: Thanks for the comments. We carefully read your comments and incorporated the changes accordingly in the paper.

The key concept of our work is about creating catalytically active nano materials starting from their liquid metal bulks and our focus is on nano-alloys. At the last part of the Results section, we present the characterization and the photocatalytic activity of the annealed samples (oxides) mainly based on two considerations: (1) They show that our strategy of processing alloys in their liquid states can both produce nano-alloy (partially oxidised) electrochemical catalysts, and also be further utilised to create oxide photocatalytic materials. (2) More importantly, we show that the Bi-Sn oxide sample prepared from the eutectic composition also possesses the highest defect intensity.

The following part is now added to the manuscript (Page 24) to address your comment: “The characterisation and the photocatalytic experiment of the annealed samples show that our strategy of processing alloys in their liquid state can both produce nano-alloy (partially oxidised) electrochemical catalysts, and also be further utilised to produce oxide photocatalytic catalysts. Also, the Raman results provide a spectroscopic proof to support our TEM microscopic defect characterisations that processing the eutectic composition leads to more intense defect formation.”

The following discussion is added (Page 21) to address your comments regarding the catalytic activity: “The ability of an electrocatalyst to produce a specific product during CO₂RR can be explained by the stabilisation of different reaction intermediates on a catalyst surface. In the case of Sn and Bi catalysts, HCOO⁻ is preferentially formed during CO₂RR [*Adv.Mater.* 2016, 28, 3423–3452]. CO₂RR on Bi and Sn catalysts proceed via the first electron transfer to the CO₂ reactant to form CO₂ anion radical (*CO₂⁻) intermediate, which is generally accepted to be the rate determining step for CO₂RR [*Adv.Mater.* 2016, 28, 3423–3452]. Subsequent proton and electron transfers lead to the formation of HCOO⁻ through either bidentate intermediate (*OCHO) or adsorbed carboxyl (*COOH) species [*Phys. Chem. Chem. Phys.* **2016**, 18, 9652; *ChemSusChem* **2017**, 10, 4342]. For the Bi-Sn nano-alloys, both

the synergistic effect between Bi and Sn on the surface electronic state and the crystallographic defects of the catalysts can affect CO₂RR. On one hand, the presence of a metal (e.g., Bi) that is more electronegative than the parent metal itself (e.g., Sn) improves the selectivity of the catalyst towards formate, since the p and d orbitals of Sn electron states can be upshifted away from the Fermi level, leading to faster electron transport to the CO₂ reactant [*Angew. Chem. Int. Ed.*, 2017, 56, 12219–12223; *J. Am. Chem. Soc.* 2017, 139, 1885-1893; *Adv. Energy Mater.* 2018, 8, 1802427]. On the other hand, defects such as grain boundaries can act as active sites for CO₂RR as a result of their favourable electronic and chemical properties [*J. Am. Chem. Soc.* 137, 4606-4609 (2015)]. These grain boundaries are reported to tune the binding energy of the CO₂RR reaction intermediates, thereby increasing the formate current density j_{HCOO^-} [*Angew. Chem., Int. Ed.* **56**, 3645-3649 (2017)]. Therefore, the observed enhanced j_{HCOO^-} of the eutectic nano-alloy sample (Fig. 6c) suggests that defects play a major role here, which matches the increased grain boundary intensity within the eutectic samples.”

We agree with you about the reliability of using HRTEM to identify point defects, so we removed the image and the corresponding discussion from the manuscript. As for your suggestion for using the EBSD method, unfortunately we do not have access to EBSD. In the past few weeks tried to communicate with many other institutions across Australia with no success. As such, to provide more evidence to support our grain size characterisations, we further conducted dark-field TEM for different Bi-Sn nano-alloy samples and also included high resolution SEM images of the bulk samples.

As shown in Fig. A1 below (dark-field TEM), the distribution of crystallites within the eutectic nano-alloys is more uniform across individual particles, indicating finer mixing of different phases and smaller grain size of the eutectic sample. Additionally, Fig. A2 below was modified and further discussion on the high-resolution SEM was added. Fig. A1 is added

to the Supplementary Information and the discussions below are added to the main text of the revised manuscript (Page 18) to address your comments.

“The distinct crystallisation behaviours between the eutectic sample and non-eutectic samples are thought to be responsible for their structural differences after solidification. The solidification of the Bi-Sn alloys is governed by the formation of nuclei and their subsequent growth, with the later process being much faster than the former³². Therefore, the solidification of the hypoeutectic ($x = 0.20$ and 0.40) and the hypereutectic ($x = 0.80$) Bi-Sn alloys start with the nucleation and growth of Sn and Bi, respectively. Since the growth of crystals takes place preferentially on crystalline faces and the Bi-Sn system has no intermetallic phase, it results in a gradual growth of Bi and Sn during the solidification that leads to the formation of larger grains of these metals for the non-eutectic alloys. For the eutectic ($x = 0.57$) alloy, the two types of nuclei form simultaneously during solidification, and the successive growth of Bi and Sn establishes the eutectic structure rapidly. As a result, the separation of the Bi and Sn phases takes place more locally, leading to smaller grain sizes and higher intensity of grain boundaries than the non-eutectics.

The above-mentioned thermodynamics of alloy solidification applies to both the Bi-Sn nano-alloys and bulk alloys, so some observations during the solidification of Bi-Sn bulk alloys can be associated to that of the nano-alloys. The zoom-in features of bulk samples after the solidification are shown in Fig. 2b. As can be seen, significant stress build-up in the bulk eutectic sample can be established from the highly distorted fibrous structures. This is in agreement with more enhanced defect formation in the eutectic nano-alloys, resulting in smaller grains in general (Fig. 5). For the non-eutectic samples, the gradual growth during solidification allows for the establishment of large rods or lamellae of Bi and Sn in bulk samples (Fig. 2b), that are relatively stress free at their grain boundaries. Similarly, for nano-alloys, this is associated with the formation of relatively larger grains of Bi and Sn (Fig. 5).”

Fig. A1 Dark-field TEM images of different $\text{Bi}_x\text{Sn}_{1-x}$ nano-alloy samples with Bi ratio x indicated.

Fig. A2 **a** SEM images and EDX element mappings showing the distribution of the Bi and Sn phases in four types of solidified structures observed in the $\text{Bi}_x\text{Sn}_{1-x}$ bulk alloys. **b** Magnified views featuring the Bi-Sn interphase regions of different solidified structures. **c** Distribution of different solidified structures in different $\text{Bi}_x\text{Sn}_{1-x}$ bulk alloys.

Reviewer #2 (Remarks to the Author):

Comment 1: The authors' revised manuscript significantly addresses this reviewer's major comments. It was good that the authors acknowledged they are working with a 3-component system (with oxygen being the third one). There are some excellent materials characterization results.

Response 1: We are very glad to read that you have seen significant improvement in our manuscript. Also, we thank you for your comments and helpful suggestions which have made the improvements possible. Our responses to your further concerns are listed below.

Comment 2: The authors make several assertions, that are not strongly justified or that are based on assumptions. In writing "...sonicating the Bi-Sn liquid alloys at a mildly elevated temperature..." the authors assert that the bulk material was melted, and that the temperature was higher than room temperature. Elsewhere they write with similar assertions like "The Bi-Sn nano-alloys prepared by liquid phase sonication..." There is no proof that the metals are in the liquified state.

Response 2: Bearing your comments in mind, we conducted further experiments to show that the samples can all be melted during the sonication process. We chose the samples with the highest melting point for demonstration, namely the $\text{Bi}_{0.80}\text{Sn}_{0.20}$ alloy sample with the DMSO group (Fig. B1a) and the Bi metal with the silicone oil sonication group (Fig. B1b), respectively. We used the same hotplate temperature $400\text{ }^{\circ}\text{C}$ (without inserting the sonicator probe). As can be seen from Fig. B1a and B1b, both bulk samples became liquid droplets under our experimental conditions, indicating the solid-to-liquid transition. The temperature of the solvent at the sample vial bottom is measured to be $187.6 \pm 0.6\text{ }^{\circ}\text{C}$ for DMSO and $284.4 \pm 0.3\text{ }^{\circ}\text{C}$ for silicone oil. Since the bottom of the vial has higher temperature than the

solvent, our configurations allow all the samples to melt. The complete melting of the samples is confirmed before applying ultrasonication during sample preparation. The details regarding the measured temperature are added to the Methods section and Fig. B1 is also added to the revised Supplementary Information to address your concern.

Fig. B1 Proof of the liquefied state of the samples. **a** The $\text{Bi}_{0.80}\text{Sn}_{0.20}$ sample in DMSO solvent. **b** The Bi metal control sample in silicone oil. The hotplate is set to 400°C and its surface temperature is measured to be $397.4 \pm 0.4^\circ\text{C}$ by an infrared thermometer.

Comment 3: They write, "Peculiar phase separation occurs during sonication that has also been shown by Tang et al.³⁸. These results imply that when the liquid bulk alloys are fragmented into nanodroplets during ultrasonication, the localised composition may be changed." The solid samples are phase-separated for most compositions. This may be the simpler reason for the observed heterogeneity in the formed nano-particles.

Response 3: Thanks for the comment. According to our observations, the phase separation in the nano-alloys share some similar mechanisms with the solid bulk samples. To highlight this fact, we added Fig. B2 as below. A full discussion regarding the association of the bulk and nano-alloys is now added to the manuscript.

Fig. B2 a SEM images and EDX element mappings showing the distribution of the Bi and Sn phases in four types of solidified structures observed in the $\text{Bi}_x\text{Sn}_{1-x}$ bulk alloys. **b** Magnified views featuring the Bi-Sn interphase regions of different solidified structures. **c** Distribution of different solidified structures in different $\text{Bi}_x\text{Sn}_{1-x}$ bulk alloys.

However, considering the differences between the nano-alloy and bulk alloy solidification, it is likely that the heterogeneity of the nano-alloys further results from: (1) the extra energy applied during the sonication, and (2) possibility of liquid and solid phase coexistence in small dimensions that has been reported previously [*Nat. Mater.* **15**, 995 (2016)].

Different phases in the bulk solid samples features micrometre separated regions (Fig. 2), while those of the nano-alloys are typically around ten nanometres (Fig. 5). Therefore, sonication and possibility of solid-liquid co-existence in a confined dimension, breaks the bulk samples and hinders the gradual growth into very large features within nanoparticles. As such, the nano-alloys constitute much more finely mixed grains.

Comment 4: The ultrasonication experiments look too difficult and dangerous for other researchers to replicate. They use a commercial probe sonicator for 2-hr experiments with 10-second on/off cycles. No temperature was recorded or reported. The hot plate apparently can be set to 400C, and they observe some DMSO evaporating during process. DMSO has bp of ~190C, indicating temperatures can reach 190C. The DMSO (and silicone oil?) vapor poses flammability and safety issues.

Response 4: Thanks for the comment. As you requested, we provided temperature details for the sonication process in the revised manuscript (also presented in the answer to 2nd Comment). In the current study, the heating conditions are selected to make sure all the samples can be melted, and the conditions are also kept the same for different samples for the purpose of comparison. In fact, some of the samples can be sonicated well below the boiling point of DMSO (e.g. $\text{Bi}_{0.57}\text{Sn}_{0.43}$, 139 °C). Besides, the experiments are conducted in a fumehood (this point is now added in the Methods). Therefore, it should not be difficult to replicate our results in standard laboratory conditions. Moreover, as listed in Table R1, there are also non-flammable liquid options to conduct similar experiments (such as silicone oil). By selecting suitable solvents, the issues such as flammability can be avoided. In the revised manuscript, we pointed out flammability of the DMSO vapor and reminded the readers that good ventilation is required. Table B1 is also added to the Supplementary Information.

Table B1. Properties of some liquids for ultrasonication

Liquid	Melting point	Boiling point	Solubility in water	Flammability
Water	0	100	-	Non-flammable
Dimethyl sulfoxide (DMSO)	19	189	Miscible	Flammable
Glycerol	17.8	290	Miscible	Flammable
Paraffin wax	37	> 370	Immiscible	Flammable
Silicone oil	< 0 (pour point)	> 300	Immiscible	Non-flammable
AlBr ₃	97.5	257	Soluble	Non-flammable
AlCl ₃ -KCl (33:67 mole %)	90		Soluble	Non-flammable

Comment 5: They cite something called "Taylor's formula," but do not make use of it. I've never heard of this equation before. The reference that they cite is the original 1934 paper by the famous Geoffrey I. Taylor, in which the capillary number (an important equation) is defined.

Response 5: Thanks for raising your concern. The equation $d \propto \sigma(x)/(\eta_c \dot{\gamma})$ is a formula that Taylor used for predicting droplet size produced by shear force. It is used in many reports since then, particularly by researchers in the fields of ultrasonics and emulsion technologies [e.g. *Ultrason. Sonochem.* 16, 721–727 (2009); *Soft Matter*, 12, 1452 (2016); *J. Phys.: Condens. Matter* 18, R635–R666 (2006)]. Two of these are now added to the paper.

In its simple form, it shows the correlation between the characterise droplet diameter d , the shear rate $\dot{\gamma}$, the viscosity of the continuous phase (solvent) η_c , and the specimen-solvent interfacial tension $\sigma(x)$. In our experiments, since the ultrasonication power, the solvent, and the heating conditions (temperature) are kept the same values for the sonication of Bi-Sn alloys, the same solvent viscosity η_c and shear rate $\dot{\gamma}$ can be assumed. Therefore, the

characterise droplet diameter d is expected to be proportional to the single variant $\sigma(x)$. Consequently, the equation shows that the decrease of particle size in the Bi-Sn nano-alloy sample as Bi ratio x increase is due to the decrease of $\sigma(x)$. Note that the composition dependence of surface tension of the Bi-Sn system is accessed in a previous work *J. Electron. Mater.* **30**, 1104-1111 (2001) and a similar dependence of interfacial tension is assumed. We carefully edited this part in the revised manuscript to ensure your comment is addressed.

Comment 6: Figure 2h showing "touch printing surface oxide layer" is not discussed well in text. Touch printing was not defined or elaborated upon.

Response 6: Thanks for your comment. The technique has been used in other liquid metal systems which are referenced in the revised manuscript. Also, details regarding the touch printing surface oxide layer is added to the Methods section as follows: "The touch-print oxide layer samples for the Raman tests were prepared by first heating the $\text{Bi}_x\text{Sn}_{1-x}$ bulk alloys on a hotplate at 400°C in atmospheric air. When melted, a heated glass slide was employed to squeeze the liquid metal drop to expose fresh liquid metal surface. A heated silicone substrate was then placed in touch with the freshly oxidised surface and lifted, after which the surface oxide layer was transferred onto the substrate."

Comment 7: Lines 385-391. The writing about the coking observation is circuitous and almost nonsensical, to the casual reader. First, carbonaceous deposits should not be called "coke". Coke is specifically formed at high-temperatures ($>500^\circ\text{C}$) in gas-phase reactions., almost graphitic in nature. Second, they write that " this is a discussion relevant to above the 139°C operating (melting) temperature for which the electrolyte should be replaced." Why would the CO_2 electrocatalytic reduction reaction be operated above 139°C ?

Response 7: We agree with you regarding this comment. We have removed this part from the revised manuscript.

Comment 8: The authors make the simplistic argument that more defects is better for higher catalytic performance. In catalysis, the choice of metals is critical for activity, because on the affinity of the metal (characterized by its d-band energy) with CO₂. The authors did not explain in manuscript why Bi-Sn nano-alloy catalysts would be active for CO₂ electrocatalytic reduction.

Response 8: We agree with you that the choice of electrode material is critical for CO₂ reduction reaction (CO₂RR) activity.

We further added the following part to the revised manuscript (Page 21) to address your concerns: “The ability of an electrocatalyst to produce a specific product during CO₂RR can be explained by the stabilisation of different reaction intermediates on a catalyst surface. In the case of Sn and Bi catalysts, HCOO⁻ is preferentially formed during CO₂RR [*Adv.Mater.* 2016, 28, 3423–3452]. CO₂RR on Bi and Sn catalysts proceed via the first electron transfer to the CO₂ reactant to form CO₂ anion radical (*CO₂⁻) intermediate, which is generally accepted to be the rate determining step for CO₂RR [*Adv.Mater.* 2016, 28, 3423–3452]. Subsequent proton and electron transfers lead to the formation of HCOO⁻ through either bidentate intermediate (*OCHO) or adsorbed carboxyl (*COOH) species [*Phys. Chem. Chem. Phys.* **2016**, 18, 9652; *ChemSusChem* **2017**, 10, 4342]. For the Bi-Sn nano-alloys, both the synergistic effect between Bi and Sn on the surface electronic state and the crystallographic defects of the catalysts can affect CO₂RR. On one hand, the presence of a metal (e.g., Bi) that is more electronegative than the parent metal itself (e.g., Sn) improves the selectivity of the catalyst towards formate, since the p and d orbitals of Sn electron states can be upshifted away from the Fermi level, leading to faster electron transport to the CO₂ reactant [*Angew.*

Chem. Int. Ed., 2017, 56, 12219–12223; *J. Am. Chem. Soc.* 2017, 139, 1885-1893; *Adv. Energy Mater.* 2018, 8, 1802427]. On the other hand, defects such as grain boundaries can act as active sites for CO₂RR as a result of their favourable electronic and chemical properties [*J. Am. Chem. Soc.* 137, 4606-4609 (2015)]. These grain boundaries are reported to tune the binding energy of the CO₂RR reaction intermediates, thereby increasing the formate current density j_{HCOO^-} [*Angew. Chem., Int. Ed.* 56, 3645-3649 (2017)]. Therefore, the observed enhanced j_{HCOO^-} of the eutectic nano-alloy sample (Fig. 6c) suggests that defects play a major role here, which matches the increased grain size distributions within the eutectic samples.”

Comment 9: Line 419. The photocatalytic results are not impressive, to this reviewer. There are plenty of properly performed photocatalytic reactions published in the literature (environmental science/engineering journals, for example), that can serve as models for how to carry out the reaction rigorously. The annealed nano-alloys have two phase Bi₂O₃ and SnO₂, as they claim. They show data that a binary oxide phase Bi₂Sn₂O₇ is found in the eutectic material – is this phase also found in the other compositions? Could the different amounts of this photoactive phase Bi₂Sn₂O₇ be the reason for the maximal activity for the eutectic material? The authors did not offer an explanation, but re-stated their observation as: "it may be a general trend for the eutectic ratio to be the optimum for developing catalysts using liquid phase ultrasonication." This was raised in my original comments. The authors gave a more detailed response in the rebuttal document, but none of this was introduced in the revised manuscript.

Response 9: Thanks for the comments. Bearing your suggestions in mind, we conducted extra XRD tests for all the annealed Bi-Sn nano-alloy samples and the results are presented in Fig. B3. The eutectic sample do show higher amount of Bi₂Sn₂O₇ than the non-eutectic samples after annealing. Therefore, the higher amount of Bi₂Sn₂O₇ phase could be possibly

associated to the higher activity of the annealed eutectic sample (all the three oxides, SnO_2 , Bi_2O_3 and $\text{Bi}_2\text{Sn}_2\text{O}_7$, are photoactive phases). Fortunately, this is not against our conclusions. Most likely, the scenario after annealing is that the binary oxide phase of $\text{Bi}_2\text{Sn}_2\text{O}_7$ is established at the grain boundaries of the Bi and Sn phases where the two initially metallic phases are in contact with each other. Therefore, it is reasonable for the eutectic sample which has the largest amount of grain boundaries to form more of this binary oxide phase. Fig. B3 is added to the Supplementary Information and the following discussion is added to the revised manuscript (Page 22) to address your comments.

“As shown in Supplementary Fig. 14, more $\text{Bi}_2\text{Sn}_2\text{O}_7$ is detected in the annealed eutectic sample. Most likely, the scenario describes this increase is that after annealing, a large amount of the binary oxide phase of $\text{Bi}_2\text{Sn}_2\text{O}_7$ is established at the intense grain boundaries of the Bi and Sn phases in the eutectic sample, where the two initially metallic phases are in contact with each other.”

Fig. B3 XRD patterns of the annealed Bi-Sn nano-alloy samples with their starting Bi ratio indicated.

Reviewer #3 (Remarks to the Author):

Comment 1: The revised version reads more correct at least, by removing previously inappropriate or even wrong statements.

Response 1: Thank you very much for your comments and suggestions which enable us to improve the manuscript.

Comment 2: On equation 1 where they tried to defend why the particle size decreases as the Bi ratio increases: they were assuming the surface tension dominates, rather than the viscosity and shear rate. This has to be justified, as otherwise the equation does not mean much.

Response 2: Thanks for the comment. In order to address the comment, we included more clarification.

Equation 1 shows the correlation between the characterise droplet diameter d , the shear rate $\dot{\gamma}$, the viscosity of the continuous phase (solvent and not the liquid alloy) η_c , and the specimen-solvent interfacial tension $\sigma(x)$.

In our experiments, since the ultrasonication power, the solvent, and the heating conditions (temperature) are kept the same for the Bi-Sn alloys, the same solvent viscosity η_c and shear rate $\dot{\gamma}$ are applied during the sonication process of all samples. Therefore, the droplet diameter d is expected to be only a function of the single variant $\sigma(x)$, the specimen-solvent interfacial tension. Considering the solvent is kept the same for all samples, the surrounding viscosity and shear rate remains the same. As such, the equation shows particle size of nano-alloy sample decreases as Bi ratio x increases only due to the decrease of $\sigma(x)$. We carefully revised this part in the revised manuscript to ensure your comment is addressed.

Comment 3: In the end, the reviewer is still confused on what happens indeed during the ultra-sonication process: so a phase separation in the liquid occurs first, and then one separated liquid phase still goes through the eutectic reaction? What drives the phase separation in the liquid state? What is the solid evidence to prove the eutectic reaction indeed occurs, other than what is shown in Fig. 3k, which is certainly not a convincing one? Note that if the liquid phase separation does happen, they cannot assume the eutectic reaction can still necessarily occur, as the composition already shifts.

Response 3: Thanks for raising your concerns. The phase separation is driven by the combined effects of solidification thermodynamics, co-existence of solid-liquid phases as has been reported previously [*Nat. Mater.* **15**, 995 (2016)], and mechanical agitation applied by sonication.

In order to answer your comment, we included several extra analyses including dark field TEM, high-resolution SEM and SEM image analysis. The discussion on dark field TEM and high-resolution SEM are further expanded in the reply to your next comment with the inclusion of the new images. The outcomes of extra SEM analysis are presented here in details.

The structures of alloys are further explored using the properties of the bulk samples to show their association to nano alloys within similar solidification thermodynamics. A statistical analysis of >10 images for each case is added to provide a systematic picture. As shown in Fig. C1, the eutectic bulk sample is dominated by highly distorted fibrous structures, which is related to the high intensity grain boundaries of the eutectic nano-alloys, since their formations both require stress build-up. The high-resolution SEM (Fig. C3b, next response) shows how the stress build up distorts the solidified crystals for eutectic alloy in comparison to other mixes. The lower grain boundary intensity (larger grain size) in the non-eutectic nano-alloys is matched by the relatively stress-free rod-like and laminar structures in the non-

eutectic bulk samples. A further discussion on this matter is added to the main text of the revised manuscript to address your concern as below (in addition to the materials in response to your next comment):

Page 6: “As shown in Fig. 2a-c, for the Bi-Sn sample with the lowest Bi ratio ($x = 0.20$), Bi forms micron/sub-micron dimensional rods (almost 100%), which discretely embedded into the Sn background phase. While with lower frequency of occurrence (16%), this discrete growth is also observed as the Bi ratio is increased to $x = 0.40$, where the Bi and Sn phases start to cut into each other, and the well-known fibrous and lamellar structures emerge³². When the Bi-Sn ratio reaches the eutectic value ($x = 0.57$), the fibrous (59%) and lamellar (41%) structures become dominant, ruling out the discrete growth regime as observed in the hypoeutectic samples. When the mixing becomes hypereutectic ($x = 0.80$), Bi grows into thick lamellae (76%) with small inter-lamella spacings.”

Page 18: “The distinct crystallisation behaviours between the eutectic sample and non-eutectic samples are thought to be responsible for their structural differences after solidification. The solidification of the Bi-Sn alloys is governed by the formation of nuclei and their subsequent growth, with the later process being much faster than the former³². Therefore, the solidification of the hypoeutectic ($x = 0.20$ and 0.40) and the hypereutectic ($x = 0.80$) Bi-Sn alloys start with the nucleation and growth of Sn and Bi, respectively. Since the growth of crystals takes place preferentially on crystalline faces and the Bi-Sn system forms no intermetallic phase, it results in a gradual growth of Bi and Sn during the solidification, which leads to the formation of larger grains hence lower grain density of these metals for the non-eutectic alloys. For the eutectic ($x = 0.57$) alloy, the two types of nuclei form simultaneously during solidification, and the successive growth of these nuclei progresses rapidly. As a result, the separation of the Bi and Sn phases takes place more locally, leading to smaller grain sizes and higher intensity of grain boundaries than the non-eutectics.

The above-mentioned thermodynamics of alloy solidification applies to both the Bi-Sn nano-alloys and bulk alloys, so some observations during the solidification of Bi-Sn bulk alloys can be associated to that of the nano-alloys. The zoom-in features of bulk samples after the solidification are shown in Fig. 2b. As can be seen, significant stress build-up in the bulk eutectic sample is established from the highly distorted fibrous structures. This is in agreement with more enhanced defect formation in the eutectic nano-alloys, resulting in smaller grains in general (Fig. 5). For the non-eutectic samples, the gradual growth during solidification allows for the establishment of large rods or lamellae of Bi and Sn in bulk samples, that are relatively stress free at their grain boundaries. Similarly, for nano-alloys, this is associated with the formation of relatively larger grains of Bi and Sn.

Different phases in the bulk solid samples features micrometre separated regions (Fig. 2), while those of the nano-alloys are typically around ten nanometres (Fig. 5). However, considering the differences between the nano-alloy and bulk alloy solidification, it is likely that the heterogeneity of the nano-alloys further results from: (1) the extra energy applied during the sonication, and (2) possibility of liquid and solid phase coexistence in small dimensions that has been reported previously⁴⁹. A such, the nano-alloys constitute much more finely mixed grains.”

Fig. C1 Distribution of different solidified structures in different $\text{Bi}_x\text{Sn}_{1-x}$ bulk alloys.

Comment 4: So the eutectic nanoalloy does not have the smallest particles size, but the smallest grain size? This is not convincing on the one hand, based simply from some TEM analysis, and difficult to be perceived on the other hand. Why is that? And more defects in the eutectic nanoalloy? Honestly, the reviewer cannot see a convincing trend here. In the end, the reviewer tends to think being eutectic or not is not the key here. The low melting point for these near-eutectic alloys is the key, which enables the ultra-sonication process. There is probably no much point to emphasize that the eutectic alloy is the magic composition. They cannot prove their nanoalloy is a eutectic nanoalloy, and they also cannot convincingly show the eutectic nanoalloy, if it is eutectic indeed, is much superior to those none-eutectic nanoalloys in terms of catalytic or photocatalytic performance.

Response 4: Thanks for the comments. We understand your point and further conducted three more characterisations of: (1) dark-field TEM of different Bi-Sn nano-alloy samples, (2) high resolution SEM images of the bulk samples, and (3) statistical image analysis of SEM images of the bulk samples (expanded in the previous response), in order to confirm the effect of grain size and also further understanding why more defects are formed. Low melting point is the key, which in fact helps in the fast growth of grains that eventually form higher stress between the grains to form more defects. Consequently, more defects result in higher catalytic activity.

In brief, dark-field TEM confirms the increase in smaller size of grains within nanoparticles. High resolution SEM shows distorted structures that suggest the stress build up during the solidification that help in the formation of more defects.

As shown in Fig. C2 below (dark-field TEM), the distribution of crystallites within the eutectic nano-alloys is more uniform across the whole particles, indicating finer mixing of different phases and smaller grain size of the eutectic sample. Fig. C2 is added to the

Supplementary Information and the discussions below are added to the main revised manuscript (Page 18) to address your comments. Additionally, Fig. C3 below was modified and further discussion on the high-resolution SEM was added. We repeat part of the materials in response to your previous comment one more time here as they cover our answer to this comment and are required for the explanation of the new figures.

“The distinct crystallisation behaviours between the eutectic sample and non-eutectic samples are thought to be responsible for their structural differences after solidification. The solidification of the Bi-Sn alloys is governed by the formation of nuclei and their subsequent growth, with the later process being much faster than the former³². Therefore, the solidification of the hypoeutectic ($x = 0.20$ and 0.40) and the hypereutectic ($x = 0.80$) Bi-Sn alloys start with the nucleation and growth of Sn and Bi, respectively. Since the growth of crystals takes place preferentially on crystalline faces and the Bi-Sn system forms no intermetallic phase, it results in a gradual growth of Bi and Sn during the solidification, which leads to the formation of larger grains hence lower grain density of these metals for the non-eutectic alloys. For the eutectic ($x = 0.57$) alloy, the two types of nuclei form simultaneously during solidification, and the successive growth of these nuclei progresses rapidly. As a result, the separation of the Bi and Sn phases takes place more locally, leading to smaller grain sizes and higher intensity of grain boundaries than the non-eutectics.

The above-mentioned thermodynamics of alloy solidification applies to both the Bi-Sn nano-alloys and bulk alloys, so some observations during the solidification of Bi-Sn bulk alloys can be associated to that of the nano-alloys. The zoom-in features of bulk samples after the solidification are shown in Fig. 2b. As can be seen, significant stress build-up in the bulk eutectic sample is established from the highly distorted fibrous structures. This is in agreement with more enhanced defect formation in the eutectic nano-alloys, resulting in smaller grains in general (Fig. 5). For the non-eutectic samples, the gradual growth during

solidification allows for the establishment of large rods or lamellae of Bi and Sn in bulk samples, that are relatively stress free at their grain boundaries. Similarly, for nano-alloys, this is associated with the formation of relatively larger grains of Bi and Sn.”

Fig. C2 Dark-field TEM images of different $\text{Bi}_x\text{Sn}_{1-x}$ nano-alloy samples with Bi ratio x indicated.

Fig. C3 **a** SEM images and EDX element mappings showing the distribution of the Bi and Sn phases in four types of solidified structures observed in the $\text{Bi}_x\text{Sn}_{1-x}$ bulk alloys. **b** Magnified views featuring the Bi-Sn inter-phase regions of different solidified structures.

REVIEWERS' COMMENTS:

Reviewer #1 (Remarks to the Author):

The authors made detailed response to the comments.

Reviewer #2 (Remarks to the Author):

This reviewer appreciates the authors' work to address the concerns. Most responses were satisfactory.

(Comment 2) Bulk heating of macro-sized metal alloys is still no compelling proof that the metals are in the liquified state, in droplet form during the sonication process. This is the basis for the perceived advantage of "eutectic alloys" (which is in title of manuscript).

(Comment 9) The photocatalytic results remain un-impressive. The revised explanation is more sensible, and it highlights a weakness of this work: that the optimum photocatalyst composition and optimum CO₂-RR catalyst composition starts from the same precursor ratio, but the final catalyst structure are very different, and the nature active sites are speculated. The word "catalysts" is the the manuscript title, which demands a certain basic level understanding of active sites. The photocatalytic results distracts from the stronger results between the CO₂-RR results and synthesized structures.

A note: the title is an oversell of the work. To this reviewer, it is unnecessarily distracting and not reflective of the results and work done: bismuth alloy, sonication, CO₂-RR, etc.

Reviewer #3 (Remarks to the Author):

The responses to the issues raised by the reviewer are reasonable. The reviewer now recommends acceptance of the current version.

REVIEWERS' COMMENTS:

Reviewer #1 (Remarks to the Author):

Comment: The authors made detailed response to the comments.

Response: The authors sincerely thank the reviewer for providing very helpful comments to improve our work.

Reviewer #2 (Remarks to the Author):

Comment: This reviewer appreciates the authors' work to address the concerns. Most responses were satisfactory.

Response: The authors are glad to see that our responses satisfactorily addressed the comments. Also, the authors sincerely thank the reviewer for the constructive comments and suggestions.

Comment: (Comment 2) Bulk heating of macro-sized metal alloys is still no compelling proof that the metals are in the liquified state, in droplet form during the sonication process. This is the basis for the perceived advantage of "eutectic alloys" (which is in title of manuscript).

Response: Thanks for raising your concern. We further conducted a control experiment to directly sonicate a solid sample without heating. As shown in Fig. R1 below, we found that the same sonication power, without heating, was not able to break the solid sample. Fig. R1 is added as Supplementary Fig. 3c to the Supplementary Information.

Fig. R1 Comparison of sonicating with and without heating to show that heating is necessary to fragment the liquid alloy into droplets.

Comment: (Comment 9) The photocatalytic results remain un-impressive. The revised explanation is more sensible, and it highlights a weakness of this work: that the optimum photocatalyst composition and optimum CO₂-RR catalyst composition starts from the same precursor ratio, but the final catalyst structure are very different, and the nature active sites are speculated. The word "catalysts" is the the manuscript title, which demands a certain basic level understanding of active sites. The photocatalytic results distracts from the stronger results between the CO₂-RR results and synthesized structures.

Response: Thanks for the comment. The main scope of our current work is to validate the feasibility of the proposed strategy to synthesize catalytic nano-alloys via processing their liquid metal bulk and to evaluate the catalytic activity arising from compositional differences. We show that the eutectic alloy results in the best catalytic outcomes, both for CO₂-RR and also for photocatalysis. This is a proof-of-concept research that offers the base for many hundreds of other alloys to be tested for their catalytic performance using the same process in the future. We would like to emphasise that we discuss the CO₂-RR and photocatalysis in

different contexts. However, we show that the eutectic alloys provide the base for synthesising the most catalytically active materials in both cases.

To further address your other comment, a brief discussion about the nature of active sites for photocatalysis was added to the photocatalysis section (page 24) as follows:

“The good correlation between the defect intensity ratio (I_D/I_A) and the rate constant k (Fig. 7e) variation as a function of x , and that both values reach maxima for the eutectic sample, indicate that the Sn-related defects may also play a role in enhancing the number of active sites for the photocatalysis. This enhanced number of active sites, together with the maximum intensity of the binary compound $\text{Bi}_2\text{Sn}_2\text{O}_7$ (Supplementary Fig. 14), can be attributed to the augmented photocatalytic effect for the annealed eutectic sample.”

Comment: A note: the title is an oversell of the work. To this reviewer, it is unnecessarily distracting and not reflective of the results and work done: bismuth alloy, sonication, CO₂-RR, etc.

Response: We would certainly appreciate the comment. However, considering that the other reviewers are happy with the title and that we feel this is the best title to describe the work, we would like to keep it unaltered.

Reviewer #3 (Remarks to the Author):

Comment: The responses to the issues raised by the reviewer are reasonable. The reviewer now recommends acceptance of the current version.

Response: The authors appreciate that our responses meet the reviewer's expectations. Additionally, the authors would like to thank the reviewer for supporting the current version of the manuscript for publication.